



# Sources and processes that control the submicron organic aerosol composition in an urban Mediterranean environment (Athens): a high temporal-resolution chemical composition measurement study CE1

**Iasonas Stavroulas**[1,2,3], **Aikaterini Bougiatioti**[1,3], **Georgios Grivas**[3], **Despina Paraskevopoulou**[3,4], **Maria Tsagkaraki**[1], **Pavlos Zarmpas**[1], **Eleni Liakakou**[3], **Evangelos Gerasopoulos**[3], **and Nikolaos Mihalopoulos**[1,3]

[1]Environmental Chemical Processes Laboratory, Department of Chemistry, University of Crete, 71003 Crete, Greece
[2]Energy Environment and Water Research Center, The Cyprus Institute, Nicosia 2121, Cyprus
[3]Institute for Environmental Research and Sustainable Development, National Observatory of Athens, Lofos Koufou, P. Penteli, 15236, Athens, Greece
[4]School of Earth and Atmospheric Sciences, Georgia Institute of Technology, Atlanta, GA 30332, USA

**Correspondence:** Aikaterini Bougiatioti (abougiat@noa.gr) and Nikolaos Mihalopoulos (nmihalo@noa.gr)

**Abstract.** TS1 TS2 Submicron aerosol chemical composition was studied during a year-long period (26 July 2016–31 July 2017) and two wintertime intensive campaigns (18 December 2013–21 February 2014 and 23 December 2015–17 February 2016), at a central site in Athens, Greece, using an Aerosol Chemical Speciation Monitor (ACSM). Concurrent measurements included a particle-into-liquid sampler (PILS-IC), a scanning mobility particle sizer (SMPS), an AE-33 Aethalometer, and ion chromatography analysis on 24 or 12 h filter samples. The aim of the study was to characterize the seasonal variability of the main submicron aerosol constituents and decipher the sources of organic aerosol (OA). Organics were found to contribute almost half of the submicron mass, with 30 min resolution concentrations during wintertime reaching up to $200\,\mu g\,m^{-3}$. During winter (all three campaigns combined), primary sources contributed about 33 % of the organic fraction, and comprised biomass burning (10 %), fossil fuel combustion (13 %), and cooking (10 %), while the remaining 67 % was attributed to secondary aerosol. The semi-volatile component of the oxidized organic aerosol (SV-OOA; 22 %) was found to be clearly linked to combustion sources, in particular biomass burning; part of the very oxidized, low-volatility component (LV-OOA; 44 %) could also be attributed to the oxidation of emissions from these primary combustion sources.

These results, based on the combined contribution of biomass burning organic aerosol (BBOA) and SV-OOA, indicate the importance of increased biomass burning in the urban environment of Athens as a result of the economic recession. During summer, when concentrations of fine aerosols are considerably lower, more than 80 % of the organic fraction is attributed to secondary aerosol (SV-OOA 31 % and LV-OOA 53 %). In contrast to winter, SV-OOA appears to result from a well-mixed type of aerosol that is linked to fast photochemical processes and the oxidation of primary traffic and biogenic emissions. Finally, LV-OOA presents a more regional character in summer, owing to the oxidation of OA over the period of a few days.

## 1 Introduction

Exposure to fine particulate matter is recognized as a leading cause of premature mortality in Europe. While the annual concentration limit may not be exceeded at the majority of regulatory monitoring stations in European countries, health effects are also expected to appear at lower concentration levels – even at levels below the WHO guideline values (EEA, 2017). Organic carbon (OC) is among the key

PM components that record the strongest associations with short-term mortality (Ito et al., 2011; Klemm et al., 2011). Moreover, short-term exposure to OC has been linked to respiratory and cardiovascular hospital admissions (Levy et al., 2012; Zanobetti et al., 2009) and pediatric asthma emergency department visits (Strickland et al., 2010). In view of the health significance of fine aerosols, the characterization of their chemical properties and short-term variability is critical, especially at the urban background level which is more relevant for average population exposure. In addition, while the majority of transformations related to particulate sulfate and nitrate have been well described, there is much progress to be made regarding the understanding of mechanisms that govern secondary organic aerosol (SOA) formation from precursors.

Therefore, the development of the aerosol mass spectrometer (AMS) technology has been an important breakthrough, facilitating the study of aerosol chemical composition, at high temporal-resolution. The ability to differentiate between primary and secondary components, based on specific markers, introduces an important advancement to organic aerosol (OA) source apportionment (Jimenez et al., 2009), which has otherwise mainly relied on a statistical approach using elemental carbon (EC) and OC thermal–optical data (EC tracer method and variants; Turpin and Huntzicker, 1995). Capitalizing on abundant spectroscopic data, PMF (positive matrix factorization) source apportionment (SA) is used to discern between various primary sources like traffic and biomass burning, and to categorize secondary aerosols depending on their degree of oxidation. The ACSM (Aerosol Chemical Speciation Monitor) is an instrument that relies on AMS technology and enables long-term routine monitoring (Ng et al., 2011 TS3).

While many relevant studies have focused on regional and rural background areas, long-term ACSM results from large European urban centers are relatively scarce. Canonaco et al. (2013) performed 1 year of measurements at an urban background site in the center of Zurich. Aurela et al. (2015) deployed an ACSM at residential, traffic, and highway sites within the Helsinki metropolitan area for a total of 5 months. Findings from 10 months of measurements at the North Kensington urban background site in London were reported by Reyes-Villegas et al. (2016). Focusing on southern European cities, long-term results are provided by the intensive ACSM campaign of Minguillon et al. (2016), at an urban background site in Barcelona. Shorter – up to 1 month – studies using the AMS have also been conducted in Barcelona (Mohr et al., 2012), Bologna (Gilardoni et al., 2016), and Marseille (El Haddad et al., 2013). In urban Athens, a 1-month AMS campaign during winter 2013 was carried out for chemical composition and OA sources (Florou et al., 2017).

The greater Athens area (GAA) appears as a challenging urban milieu for the study of aerosol dynamics, as it combines a large population (about 4 million) and intense primary emissions, with complex topography and meteorology, which lead to high levels of atmospheric pollutants and significantly deteriorate the air quality (Kanakidou et al., 2011; Pateraki et al., 2014). However, the characteristics and related processes of SOAs, in the long term, have received limited attention up until this point (Grivas et al., 2012; Paraskevopoulou et al., 2014). Moreover, since 2013, due to the economic recession in Greece, primary and secondary precursor emissions have become altered and intensified, as residents have switched from fossil fuel combustion to the uncontrolled burning of wood and biomass for space heating (Saffari et al., 2013; Fourtziou et al., 2017; Gratsea et al., 2017). Existing measurements of aerosol chemical composition in Athens have mainly been performed using filter sampling (Theodosi et al., 2011, 2018 TS4; Paraskevopoulou et al., 2014) and have indicated the importance of fine OAs. In this study we present, for the first time, long-term results regarding the sources of submicron OAs in Athens from high temporal-resolution measurements during a year-long period, complemented by two intensive winter campaigns. For the collection of data, we deployed an Aerosol Chemical Speciation Monitor (ACSM) in addition to a particle-into-liquid sampler coupled with ion chromatography (PILS-IC) and an AE-33 Aethalometer, while also conducting auxiliary aerosol (filter-based) and gas phase measurements. The main objectives of this study were (i) to characterize submicron aerosol and its variability using high temporal-resolution measurements, (ii) to quantify the sources of OAs and their seasonal variability (via PMF analysis), and (iii) to study the year-to-year changes of aerosol sources during wintertime.

## 2    Experimental methods

### 2.1    Sampling site and period

The measurements exploited in this study were conducted, at the urban background site of the National Observatory of Athens (NOA) at Thissio (37.97° N, 23.72° E), as representative of the mean population exposure over the Athens metropolitan area (Fourtziou et al., 2017). The site is located at an elevation of 105 m a. s. l., in a moderately populated area, where the influence of direct local emissions is limited.

The measurement period spanned an entire year, from July 2016 to July 2017. Additionally, two intensive winter campaigns took place at the same site, the first from mid-December 2013 to mid-February 2014 and the second from 23 December 2015 to 17 February 2016. These intensive campaigns aimed at studying the year-to-year variability and impact of biomass burning on the air quality of the city of Athens during wintertime.

### 2.2    Instruments and methods

Measurements were performed with an Aerosol Chemical Speciation Monitor (ACSM) from Aerodyne Research Inc.

(Ng et al. 2011a), which measured the non-refractory $PM_1$ (NR-$PM_1$) chemical composition in near real-time (30 min temporal resolution). The instrument sampled through a BGI Inc. SCC 1.197 sharp cut cyclone operated at $3\,L\,min^{-1}$, yielding a cut-off diameter of approximately $2\,\mu m$. Practically, the ACSM operates following a similar principle to the aerosol mass spectrometer (AMS) (Jayne et al., 2000) where ambient air is drawn through a critical orifice to a particle focusing aerodynamic lens; the resulting particle beam is flash-vaporized at $600\,°C$, ionized via electron impact ionization, and guided through a quadrupole mass spectrometer. Ammonium nitrate and ammonium sulfate calibrations were performed prior to the ACSM's deployment at the site for the 2016–2017 period, and the response factor (RF) for nitrate along with the relative ionization efficiencies (RIEs) for ammonium and sulfate were determined. For the 2013–2014 and 2015–2016 intensive winter campaigns ammonium nitrate calibrations were performed and the RIE for sulfate was determined according to the fitting approach proposed by Budisulistiorini et al. (2014). Values are presented in Table S1 TS5 in the Supplement. The detection limits for the ACSM provided by Ng et al. (2011a) are as follows: $0.284\,\mu g\,m^{-3}$ for ammonium, $0.148\,\mu g\,m^{-3}$ for organics, $0.024\,\mu g\,m^{-3}$ for sulfate, $0.012\,\mu g\,m^{-3}$ for nitrate, and $0.011\,\mu g\,m^{-3}$ for chloride. Mass concentrations are calculated using a chemical composition dependent collection efficiency (Middlebrook et al., 2012; Fig. S1 in the Supplement).

Parallel measurements were performed for biomass burning identification and for quality control purposes. In this context, a Metrohm ADI 2081 particle-into-liquid sampler (PILS; Orsini et al., 2003) coupled with ion chromatography (Dionex ICS-1500) was used, which sampled ambient air from a different, but adjacent to the ACSM's, $PM_1$ inlet. Two denuders were placed inline, upstream of the instrument, in order to remove gas phase species (e.g., $NH_3$, $HNO_3$, and $SO_2$). The ion chromatograph was set to measure cations such as ammonium and potassium at a time resolution of 15 min. The resulting concentrations from the ACSM were tested against filter measurements and the concentrations provided by the PILS. For the PILS, the detection limit was calculated at 1 ppb for $Na^+$ and $NH_4^+$ and 2 ppb for $K^+$. Non-sea-salt $K^+$ (nss-$K^+$) concentrations were calculated using the $Na^+$ concentrations and the $Na^+/K^+$ ratio in seawater as a reference (Sciare et al., 2005). The concentrations reported were blank corrected.

Furthermore, filter sampling was also conducted in parallel at Thissio station. $PM_{2.5}$ aerosol samples were collected on quartz-fiber filters (Flex Tissuquartz, 2500 QAT-UP 47 mm, Pall), on a daily basis, while during the winter periods the sampling frequency was set to 12 h. A dichotomous Partisol sampler 2025 (Ruprecht & Patashnick) was used at a flow rate of $16.7\,L\,min^{-1}$. The samples were analyzed for organic and elemental carbon (OC, EC) with the thermal–optical transmission technique, using a Sunset Laboratories OC/EC analyzer and applying the EUSAAR-2 protocol (Cavalli et al., 2010). Filters where also analyzed for the determination of the main ionic species using ion chromatography as described in Paraskevopoulou et al. (2014).

Two different absorption photometers were monitoring black carbon (BC) concentrations. A 7-wavelength Magee Scientific AE-42 portable Aethalometer was used for the 2013–2014 and 2015–2016 winter campaigns, which provided 5 min resolution measurements. For the year-long period a dual spot, 7-wavelength Magee Scientific AE-33 Aethalometer (Drinovec et al., 2015) was used, operating at a 1 min resolution and a $5\,L\,min^{-1}$ flow rate. Standard gas analyzers for $O_3$ (Thermo Electron Co., Model 49i), CO, $SO_2$, and $NO_x$ (HORIBA, 360 series) in addition to a scanning mobility particle sizer for $PM_1$ size distributions (SMPS 3034, TSI Inc.), measuring in the 10.4–469.8 nm size range, were also operating at the sampling site. Wavelength dependent source apportionment of the BC load was performed by the AE-33 Aethalometer, based on the approach from Sandradewi et al. (2008), providing a fossil fuel ($BC_{ff}$) and a wood combustion ($BC_{wb}$) component. The default absorption Ångström exponents of 1 for fossil fuel combustion and 2 for pure wood burning, as incorporated in the AE-33 software, were used, which were very close to the respective values of 0.9 and 2 used at a suburban site in Athens (Kalogridis et al., 2018). Meteorological parameters for the study were taken from the actinometric meteorological station of NOA, at Thissio (Kazadzis et al., 2018) (Fig. S2). All measurements were averaged to 1 h intervals in order to synchronize the different datasets.

The bivariate wind speed–direction plotting methodology developed by Carslaw and Ropkins (2012) in the "openair" R-package, was used for the identification of source areas, as incorporated in the Zefir Igor Pro-based tool (Petit et al., 2017). Four-day back trajectories were also calculated using the HYbrid Single-Particle Lagrangian Integrated Trajectory (HYSPLIT_4) model (Draxler and Hess, 1998 TS6) developed by the Air Resources Laboratory (ARL/NOAA), and 1° GDAS (NCEP) meteorological data. Trajectories were computed every 3 h, for air masses arriving at Athens at a height of 1000 m. The selected height is considered suitable to capture transport at a representative upper limit of the boundary layer in Athens (Markou and Kassomenos, 2010). Trajectory clustering was performed using the TrajStat plug-in (Sirois and Bottenheim, 1995; Wang et al., 2009) of the MeteoInfo GIS software. The change of the total space variance for a decreasing number of clusters was examined as a criterion for cluster number selection. The analysis was performed separately for summer and winter, resulting in five clusters for each period.

## 2.3 Source apportionment of the submicron organic fraction using PMF analysis

### 2.3.1 PMF strategy

Positive matrix factorization (Paatero and Tapper, 1994) was performed on the organic mass spectra obtained from the ACSM. The graphic interface SoFi (Source Finder) version 6.1, developed at the Paul Scherrer Institute (PSI) in Zurich (Canonaco et al., 2013), was used. SoFi implements the multi-linear engine algorithm ME-2 (Paatero and Hopke, 2003), analyzing the acquired mass spectral time series matrix into a linear combination of factor profiles (FP) and time series sub-matrices. A detailed description of the method can be found in the studies referenced above.

For our datasets only $m/z \leq 125$ were used in order to avoid interference from the naphthalene signal ($m/z$ 127, 128, and 129). Weak signals, with a signal-to-noise ratio ($S/N$) below 0.2 were down-weighted by a factor of 10, and those with a $S/N$ between 0.2 and 1 were down-weighted by a factor of 2 (Ulbrich et al., 2009), using the built in utilities of the SoFi toolkit.

The input organics and the organics' error matrices are automatically derived from the ACSM data analysis software. Several model runs were performed, with and without applying constraints to the FPs derived, using the $\alpha$-value approach (Canonaco et al., 2013, 2015) and following the methodology proposed by Crippa et al. (2014). The $\alpha$ value ranges between 0 and 1 and is a measure of how much the resulting FPs are allowed to vary from the constraints CE3 . Initially, unconstrained PMF runs provided insight into the potential number and type of factors. For the following steps, reference factor profiles (RFPs) were introduced in order to constrain primary OA factors, (i) first for hydrocarbon-like organic aerosol (HOA), (ii) then for both HOA and BBOA, and (iii) finally for HOA, BBOA, and cooking-like organic aerosol (COA) CE4 . Potential FPs for SOAs were left unconstrained. A thorough discussion regarding the choice and representativeness of the RFPs used can be found in Sect. S4.1 of the Supplement. Each factor was constrained using different $\alpha$ values within the limits suggested by Crippa et al. (2014). Next, the model's residuals, for each different model setup, were analyzed in search of structures that could indicate underestimation or overestimation of the number of separated factors. Stability of factors for different model seeds and correlations of the obtained FP spectra with FPs reported in similar environments and conditions were examined (Sect. S4.8). Finally, correlations of the time series of the selected optimal solutions to both gas phase and particulate independent measurements such as BC, $BC_{ff}$, $BC_{wb}$, CO, nss-$K^+$, $NO_3^-$, $SO_4^{2-}$, CE5 and $NH_4^+$ were examined to solidify the selection (Sect. S4.9).

The year-long data series was divided into a cold period, from November 2016 to March 2017 and a warm period consisting of two sub-periods from August to September 2016

and from May to July 2017 which were treated separately. According to studies on the climatology of southern Greece, the transient period (spring and fall seasons) in Athens does not exceed 60 days on average (Argyriou et al., 2004), and mainly spans the months of April and October – which were excluded from the seasonal analysis. The two wintertime campaigns of 2013–2014 and 2015–2016 were also treated separately.

The coefficient of determination $r^2$ for simple linear regression is used as a metric for all comparisons, e.g., both the affinity of the FPs obtained to spectra from the literature and the correlation of the respective factor time series with independent measurements.

### 2.3.2 Choosing the optimal configuration

The presentation of, and discussion of the optimal configuration chosen for the ME-2 model, as well as results from each step of the implemented strategy described above, followed by a sensitivity analysis of the $\alpha$ value influence on the obtained factors, can be found in Sect. S4. In brief, for the cold period and the two wintertime intensive campaigns, constraining three factors, namely HOA, BBOA, and COA, and leaving two unconstrained SOA factors, produced a solution that is characterized by minimal seed variability and model residual structures, while the FPs, the time series, the relative contribution, and the diurnal variability of the factors appear to be environmentally relevant, resembling previously proposed solutions for the region (Kostenidou et al., 2015; Florou et al., 2017). Leaving factors unconstrained leads to unstable model behavior such as diurnal residual structures for key variables (e.g., alkyl fragments like $m/z = 55$ or 57) and large FP variability for different model seed runs. Furthermore, deconvolved spectra were missing expected variable contributions in profiles such as BBOA (very low $m/z = 41$ and 43 relative contributions), while the COA factor was dominated by the $CO_2^+$ fragment at $m/z = 44$. Configuring less or more than five factor solutions, either resulted in an even more pronounced residual diurnal cycle, pointing to poor factor separation, or in splitting behavior and resulting factors which were environmentally irrelevant.

Conversely, constraining two factors during the warm period, namely HOA and COA, and leaving two unconstrained SOA factors was found to be the solution that exhibited higher relevance while also being robust and close to previous knowledge related to OA in the GAA. A BBOA factor could not be identified during the warm periods, as the contribution of the marker fragments for biomass burning, $m/z = 60$ and $m/z = 73$, are almost absent in these periods dataset. The COA factor is present in all of the studied periods, validated following the approach of Mohr et al. (2012) (Fig. S9), and emerged in all of the steps (unconstrained and constrained runs) of the strategy implemented (Figs. S5–S8 and related discussion in Sect. S4).

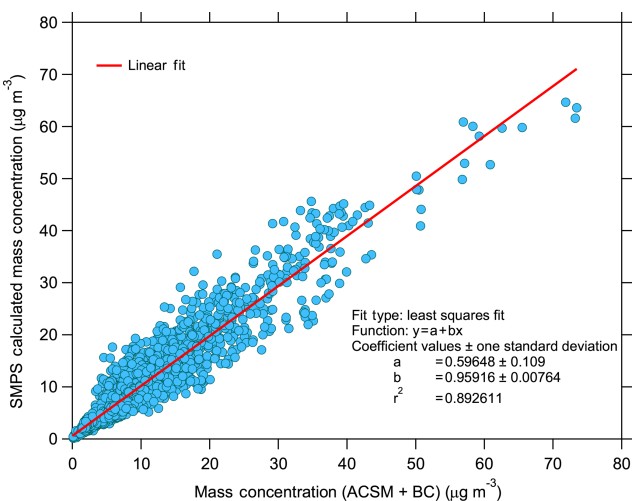

**Figure 1.** Correlation between ACSM + BC vs. SMPS-derived 1 h averaged mass concentrations for the 2016–2017 measurement period.

## 3 Results and discussion

### 3.1 Comparison of ACSM data with ancillary measurements

As an initial quality control/quality assurance of the ACSM data, the ammonium concentrations are compared to the respective values derived from the PILS, on an hourly basis for winter 2016–2017. A good agreement is found ($r^2 = 0.80$ and slope of 0.82). The sulfate and nitrate concentrations for the winter 2016–2017 period are compared to the respective values from the ion chromatography analysis (PM$_{2.5}$ filters) on a daily basis ($r^2 = 0.75$ and slope of 0.81 and $r^2 = 0.78$ and slope of 0.95, respectively). The concentrations of organics are compared to the OC concentrations of the PM$_{2.5}$ filters. An excellent agreement is found ($r^2 = 0.93$, slope of 1.59) with the slope being close to values reported for urban areas (Petit et al., 2015) and OM : OC calculations from AMS measurements in polluted environments (Saarikoski et al., 2012). The results from the aforementioned comparisons are provided in the Supplement (Sect. TS7 S3).

During the intensive winter 2015–2016 campaign, the concentrations of the ACSM components are compared to those determined from ion chromatography, based on concurrent filter samples collected at the same site, twice per day, (06:00–18:00 and 18:00–06:00 LT – local time). Results indicate an excellent agreement for sulfate ($r^2 = 0.88$, slope of 1.0), ammonium ($r^2 = 0.82$, slope of 1.06), and nitrate ($r^2 = 0.88$, slope of 1.12) (Fig. S4). During the intensive winter 2013–2014 campaign, the ammonium concentrations from the ACSM showed significant correlation with the respective concentrations from the PILS instrument ($r^2 = 0.80$, slope of 0.81).

Finally, the sum of the ACSM component concentrations plus BC, measured using the 7-wavelength Aethalometer, was compared with the mass concentrations determined by the SMPS since February 2017 at Thissio. The density used to convert volume distributions, and consequently the volume concentrations of spherical particles to mass concentrations, was obtained by applying the methodology of Bougiatioti et al. (2014) assuming that the aerosol PM$_1$ population was dominated by ammonium sulfate and organics and calculating the respective mass fractions time series based on the ACSM measurements. A density of 1.77 g cm$^{-3}$ was used for ammonium sulfate, and a value of 1.3 g cm$^{-3}$ was utilized for organics (Florou et al., 2017). The results obtained using a chemical dependent collection efficiency to determine the ACSM derived mass concentrations are portrayed in Fig. 1 and indicate excellent correlation ($r^2 = 0.89$), a slope of 0.96, and an intercept of 0.60.

### 3.2 PM$_1$ average chemical composition and temporal variability

#### 3.2.1 Chemical composition and characteristics

The time series of the main submicron aerosol components measured using the ACSM and the black carbon concentrations are presented in Fig. 2a (one complete year period). The period average cumulative concentration of the ACSM components and BC was $12.4 \pm 12.5$ µg m$^{-3}$. The highest concentrations were measured during winter (average $16.1 \pm 19.5$ µg m$^{-3}$) and the lowest during summer (average $10.3 \pm 5.6$ µg m$^{-3}$). On an annual basis, the most abundant component was OA, followed by sulfate, which contributed 44.5 % and 27.8 % to the total submicron mass, respectively, while the BC contribution was calculated at 15.1 %, ammonium at 7.9 %, and nitrate at 4.3 %. In Fig. 2b and c the respective time series of the main submicron aerosol components during the two intensive 2-month winter campaigns are presented. During winter 2013–2014 the average mass concentration of the ACSM components (plus BC concentrations) was $24.5 \pm 24.7$ µg m$^{-3}$, with organics and BC contributing 55.6 % and 14.6 % to the total submicron mass, respectively, followed by sulfate (13.6 %). During winter 2015–2016 the average concentration was $21.2 \pm 27.4$ µg m$^{-3}$, with organics and BC contributing 51.6 % and 15.2 % to the total submicron mass, respectively, followed by sulfate (14.8 %), nitrate (6.5 %), and ammonium (6.7 %). It is clearly deduced that during the last winters CE6, organics constitute half or even more of the total PM$_1$ mass, sulfate around 20 %, and BC around 14 %.

The other striking feature is that during wintertime, PM$_1$ concentration spikes can reach up to 220 µg m$^{-3}$ hourly values, with organics constituting most of the mass. Maxima are recorded during nighttime and mostly during meteorological conditions that favor pollutant emission and accumulation, such as low wind speed and low temperature (Fourtziou et

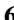

**Figure 2.** Time series of the main submicron aerosol components. **(a)** The 1-year period starting on 26 July 2016 and ending on 31 July 2017, **(b)** the 2013–2014 winter campaign (18 December–21 February), and **(c)** the 2015–2016 winter campaign (23 December–17 February).

al., 2017). On average, 8 such incidents occur each winter (10 in 2013–2014, 7 in 2015–2016, and 7 in 2016–2017), with recorded organic levels higher than $100\,\mu g\,m^{-3}$. To our knowledge, such levels are the highest reported for Europe during wintertime and highlight the strong impact of local emissions – especially those related to heating/wood burn-ing (see below) – on the levels of organics and consequently PM$_1$. Similar maxima to those observed in this study are also reported by Florou et al. (2017; at the same site from 10 January until 9 February 2013), where organics concentration alone reached up to $125\,\mu g\,m^{-3}$ and maxima of $8\,\mu g\,m^{-3}$ for BC and up to $5\,\mu g\,m^{-3}$ for nitrate were recorded. Similarly,

wintertime pollution events with an increased local character and elevated concentrations of organics (up to around 100 µg m$^{-3}$, average of 22.6 µg m$^{-3}$) were reported at a regional background site, just outside of Paris, during February 2012 (Petit et al., 2015).

### 3.2.2 Seasonal variability

The seasonal variability of the main measured species, along with the average PM$_1$ concentration (µg m$^{-3}$), as calculated from the ACSM + BC measurements is shown in Fig. 3 and the basic statistics are included in Table 1. Organics contribute 46 % to the total submicron aerosol mass in summer, followed by sulfate (30.5 %), BC (12.6 %), ammonium (8.3 %), and nitrate (2.6 %), while in winter, organics and sulfate contribute 48.1 % and 23.2 %, respectively, followed by BC (14.7 %), ammonium (6.9 %), and nitrate (6.3 %).

The mass concentrations of organics, nitrate, chloride, and BC exhibit a clear annual cycle, with a minimum during summer and a maximum in winter. This pattern seems to be due to a combination of three simultaneous processes. Firstly, the additional primary emissions from domestic heating play an important role, as is evident from the largely elevated concentrations of organics and BC, which are emitted by central heating systems and fireplaces during winter. Secondly, the decreased planetary boundary layer (PBL) depth during winter may influence observed pollutant patterns: according to Kassomenos et al. (1995) and Alexiou et al. (2018), the daytime PBL depth shows a clear annual cycle, with maxima during the warm months (June to September) and a 2-fold decrease during wintertime. Finally, the effect of temperature on the partitioning of semi-volatile inorganics and organics can also contribute to the processes leading to the observed pattern. In support of the above, a larger standard deviation is found in winter, which demonstrates the frequency and magnitude of the pollution events observed due to the increased need for heating (Fourtziou et al., 2017). Independently of the year, it can be seen that winter concentrations of organics, nitrate, chloride, and BC are very similar and are more than twice the respective values of these species during the other seasons (Table 1).

The concentrations of organics are consistently high during all of the winters studied (from December to February), while the higher nitrate values, which exhibit a similar trend to organics and BC, can be attributed to the combination of lower temperatures during nighttime and the increase of combustion sources; this leads to reduced acidity and results in the favorable partitioning of nitrate in the aerosol phase (Park et al., 2005; Mariani and de Mello, 2007; Guo et al., 2016). Ammonium and sulfate exhibit the opposite seasonal cycle, with maximum values in summer and minimum values during winter and spring. The higher summer sulfate levels are the result of enhanced photochemistry associated with more intense insolation, combined with less precipitation, favoring the regional transport of polluted air masses (Cusack

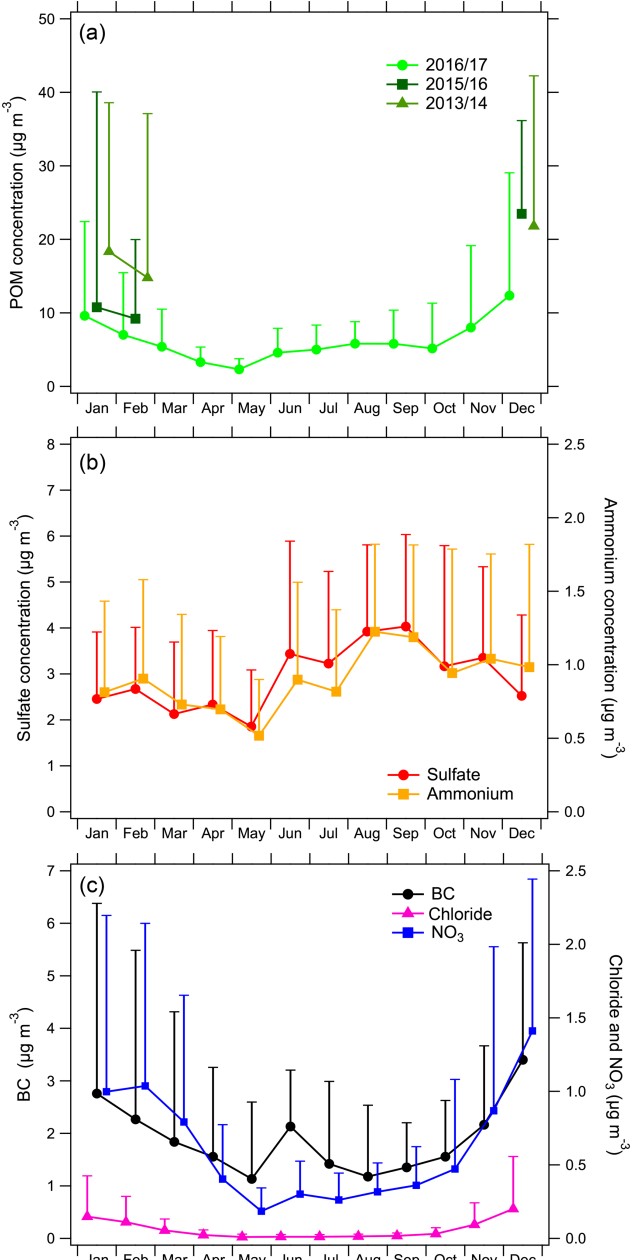

**Figure 3.** Monthly average concentrations of the main aerosol constituents. Organics are shown in **(a)** for the 2016–2017 period as well as the 2013–2014 and 2015–2016 winter periods, while sulfate and ammonium are shown in **(b)** and BC, nitrate, and chloride are shown in **(c)** for the 2016–2017 period. The standard deviation is also depicted (error bars; only the positive part is shown for the sake of clarity).

et al., 2012). The seasonal variation of concentrations is in agreement with observations from Athens made during prior long-term measurement campaigns based on the analysis of daily filter samples (Theodosi et al., 2011; Paraskevopoulou et al., 2014, 2015).

**Table 1.** Seasonal average concentrations ± standard deviation (range) and total mass of the main submicron aerosol components for the 1-year study period and the two winter campaigns.

| | March–April–May 2017 | July–August 2016 & June–July 2017 | September–October–November 2016 | December–January–February 2016–2017 | December–January–February 2013–2014 | December–January–February 2015–2016 |
|---|---|---|---|---|---|---|
| Organics | 3.3 ± 3.0 (0.3–31.3) | 5.4 ± 3.4 (0.3–41.9) | 6.1 ± 7.5 (0.1–98.2) | 9.0 ± 13.4 (0.2–153.9) | 18 ± 24.4 (0.4–212.2) | 12.4 ± 19.9 (0.7–1150.5) |
| Ammonium | 0.6 ± 0.5 (0.4–3.1) | 1.0 ± 0.6 (0.2–4.1) | 1.0 ± 0.7 (0.4–5.7) | 0.9 ± 0.7 (0.2–5.7) | 1.8 ± 1.2 (0.2–9.1) | 1.1 ± 1 (0.3–6.7) |
| Sulfate | 2.1 ± 1.5 (0.2–10.1) | 3.6 ± 2.1 (0.3–14.9) | 3.5 ± 2.3 (0.1–17.1) | 2.5 ± 1.5 (0.1–11.7) | 2.6 ± 1.4 (0.4–13.9) | 2.2 ± 1.7 (0.4–10.3) |
| Nitrate | 0.4 ± 0.5 (0.05–5.4) | 0.3 ± 0.2 (0.01–1.5) | 0.5 ± 0.7 (0.1–6.9) | 1.2 ± 1.5 (0.05–12.1) | 2.6 ± 2.4 (0.09–18.3) | 1.5 ± 1.4 (0.07–16) |
| Chloride | 0.02 ± 0.05 (0–0.8) | 0.02 ± 0.02 (0.04–0.2) | 0.04 ± 0.09 (0.07–2.0) | 0.15 ± 0.3 (0–3.5) | 0.16 ± 0.24 (0.09–8.1) | 0.12 ± 0.24 (0–2.6) |
| BC | 1.5 ± 1.4 (0.1–14.6) | 1.2 ± 0.8 (0.2–10.5) | 1.7 ± 1.6 (0.1–12.4) | 2.4 ± 3.4 (0.1–29.6) | 2.7 ± 3.2 (0.2–26.8) | 3.4 ± 4.6 (0.2–32.3) |
| PM₁ CE7 | 8.9 ± 6.1 (0.6–42.4) | 10.3 ± 5.6 (0.5–52.2) | 13 ± 11.1 (0.9–115.5) | 16.1 ± 19.5 (0.8–185.8) | 24.5 ± 24.7 (1.4–227.2) | 21.2 ± 27.4 (1.7–215.3) |

### 3.2.3   Diurnal variability

When investigating the diurnal patterns of the measured species (Fig. 4), it was observed that ammonium and sulfate did not exhibit significant variability during wintertime, which was due to the regional character of ammonium sulfate. In order to quantify the extent of this variability we calculated the normalized diurnal pattern by dividing each hourly value by the respective species' daily mean concentration. More specifically, sulfate varies by 13 % around the mean value while ammonium varies by 40 %. Conversely, organics, BC, and nitrate vary significantly during the day (183 %, 79.8 %, and 110 %, respectively). These species clearly double their concentrations during nighttime, due to the additional primary emissions. Furthermore, BC also exhibits a second maximum during the early morning hours, which can be attributed to the primary emissions during the morning traffic rush hours.

During summer, all concentrations are significantly lower, especially organics (note the scale change in Fig. 4 CE8) which exhibit a 5-fold decrease of their mean maximum concentration during nighttime. Normalizing the diurnal cycles, as mentioned above, reveals a much less pronounced variability for organics (65 %), implying a more regional character, while BC and nitrate exhibit the highest variability (67.7 % and 77 %, respectively) in accordance with their local nature. The nighttime maxima of BC vanish, while nitrate shows much lower concentrations, due to nitrate partitioning between the gas and aerosol phase, favoring the vaporization of ammonium nitrate. BC still only exhibits one maximum during the early morning hours owing to traffic emissions. The ammonium and sulfate diurnal profile follows expected photochemistry patterns, with peaking concentra-

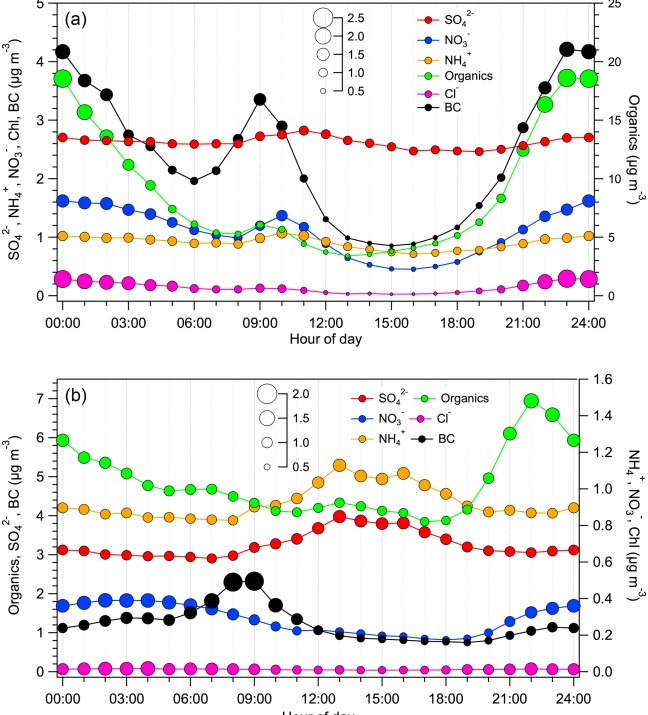

**Figure 4.** Average daily cycle of the main submicron aerosol constituents for the cold period 2016–2017 **(a)** and the warm period of 2017 **(b)**. The size of the markers indicates the normalized values relative to each species' daily mean value.

tions around 14:00 LT (UTC + 02:00); this is consistent with secondary aerosol formation and increased vertical mixing with regional aerosol from aloft due to the evolution of the

convective boundary layer which exhibits a bell shaped diurnal structure ranging from a few hundred meters to above one kilometer, with maximum heights during early afternoon (Asimakopoulos et al., 2004; Tombrou et al., 2007). Finally, the concentrations of organics are somewhat higher during the early nighttime which could possibly be associated with either local or regional biogenic/vegetation sources that produce volatile compounds and condense on the particulate phase during nighttime when temperatures are lower, as is further elaborated on during the source apportionment results discussion in Sect. 3.3. Furthermore, the variation of organics also follows the late afternoon peak observed for ammonium and sulfate. Condensation of the particulate phase could apply for nitrate as well, as this species also exhibits higher concentrations during nighttime (almost double).

### 3.3 Source apportionment of organic aerosol

*The warm period*: in this period, the selected solution stems from a two factor constrained run (HOA using $\alpha = 0.05$ and COA using $\alpha = 0.1$) and consists of four factors: HOA, COA, SV-OOA, and LV-OOA. As previously mentioned, the two summer periods were treated separately, but the spectra derived were almost identical ($r^2$ ranging from 0.98 to 0.99). The time series of the four identified sources during summer 2017 is shown in Fig. 5 along with their diurnal variability and the respective average hourly contribution. The mass spectra of the selected solution are also provided in the Supplement (Fig. S12). No primary biomass burning aerosol could be identified, which is justified by the absence of fresh emissions over the city center during the warm period. In the summer periods HOA makes up 4.3 % of the total organic fraction, while COA comprises around 10 % on average (7.3 % and 11.3 % for 2016 and 2017, respectively). In summer 2016 SV-OOA made up 32 %, while the remaining 56 % was LV-OOA. In summer 2017, SV-OOA contributed 34.6 % to the total organic fraction, while LV-OOA made up 49.7 %. The dominance of the secondary influence (SV-OOA and LV-OOA) is apparent and accounts for the majority of the OA. This finding is in accordance with Kostenidou et al. (2015), who reported that 65 % of the sampled aerosol during summer could be attributed to SOA (SV-OOA and LV-OOA) at a suburban site in Athens.

A comparison of the derived FPs against mass spectra from the literature is shown in Figs. S15–S19 in the Supplement. COA FP exhibits excellent correlation with spectra obtained during previous studies in the city (Florou et al., 2017; Kostenidou et al., 2015) as well as with spectra obtained in laboratory experiments investigating fresh OA emissions from meat charbroiling (Kaltsonoudis et al., 2017). When calculating the O : C ratio in COA following the study of Canagaratna et al. (2015) we find a ratio of 0.19, which is comparable with the value of 0.24 obtained for COA during summer at a suburban site in Athens (Kostenidou et al., 2015).

The HOA FP exhibits excellent correlation with spectra from the literature measured in cities located within the Mediterranean environment (Florou et al., 2017; Kostenidou et al., 2015; Gilardoni et al., 2016) as well as in other environmental and socioeconomic settings (Crippa et al., 2013; Lanz et al., 2009 TS8). According to Fig. S18, where the affinity of SV-OOA with literature spectra is assessed, some assumptions could be made regarding the origin of this factor: its similarity to isoprene-epoxydiol organic aerosol (IEPOX–OA), which is the oxidation product of isoprene, could denote a possible link between SV-OOA and biogenic aerosol. This association is further strengthened by considering the excellent correlation with SOA from biogenic precursors, such as $\alpha$- and $\beta$-pinene reported by Bahreini et al. (2005) ($r^2$ of 0.86 and 0.89, respectively). These precursors are found to exhibit maxima during nighttime (Harrison et al., 2001; Li et al., 2018; Hatch et al., 2011) which coincides with the diurnal behavior of SV-OOA in this study. Conversely, a comparison of the derived SV-OOA with SOA from diesel exhaust after 4 h of photochemical ageing (Sage et al., 2008) yields an $r^2$ of 0.89. Finally, SV-OOA exhibits the lowest correlations with the mass spectrum from aged OA emissions from meat charbroiling (Kaltsonoudis et al., 2017). The above-mentioned comparisons with literature FPs provide some indication that SV-OOA could be linked to SOA formation from the oxidation of volatile organic compounds (VOCs) from both biogenic and traffic sources during summer, and is not linked to the oxidation of primary COA. The low volatility component derived exhibits an excellent correlation with the very oxidized regional OOA found in the area (Bougiatioti et al., 2014) and a good correlation with deconvolved OOA factors from previous studies in Athens (Florou et al., 2017; Kostenidou et al., 2015). When calculating the elemental ratios based on the study of Canagaratna et al. (2015), the O : C ratio for LV-OOA is 1.2, which is identical to the value of OOA obtained at Finokalia (Bougiatioti et al., 2014).

In terms of comparison with independent measurements, HOA exhibits good correlation with nitrate ($r^2 = 0.62$) as well as with $BC_{ff}$ ($r^2 = 0.63$), while COA, as expected, shows poor correlation with CO ($r^2 = 0.33$) and nitrate ($r^2 = 0.36$). SV-OOA is highly correlated with nitrate ($r^2 = 0.86$), implying common mechanisms in their variability, which is possibly linked with the partitioning between the gas and particulate phases. The poor correlation with CO ($r^2 = 0.4$) and BC ($r^2 = 0.35$) implies that SV-OOA may, to some extent, partially originate from a combustion source. LV-OOA shows good correlation with sulfate ($r^2 = 0.62$) and ammonium ($r^2 = 0.63$), which is consistent with the regional character of this factor. Results from the trajectory cluster analysis (Fig. S21) show that enhanced LV-OOA levels are related to air masses originating from eastern Europe and the Black Sea region, which have both been identified as the main areas of influence for secondary aerosols that are regionally processed and transported to Athens (Gerasopoulos et al., 2011; Grivas et al., 2018). The regional character of LV-OOA is

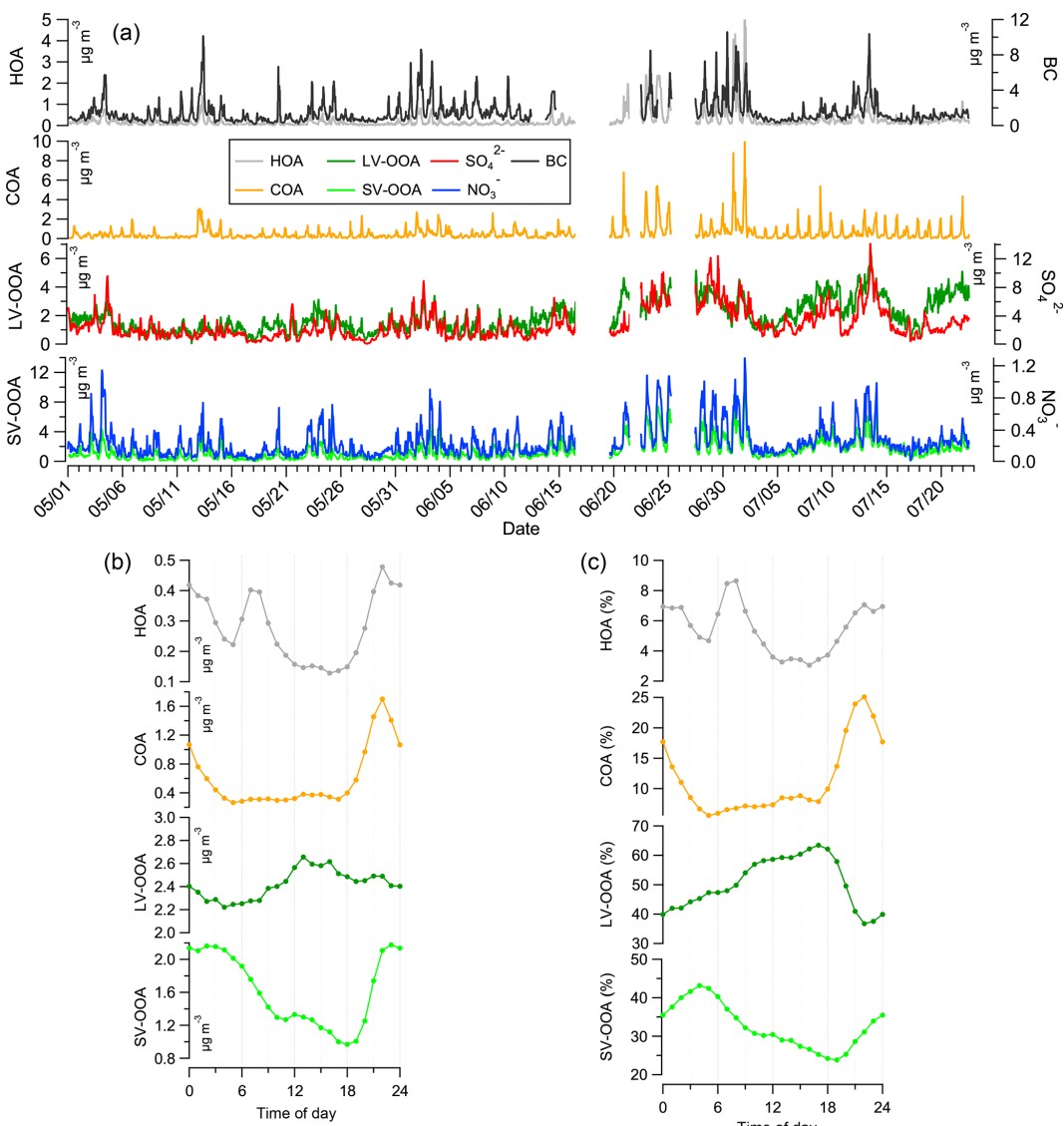

**Figure 5.** Time series of the contribution of the different factors identified by PMF between 1 May and 31 July 2017 **(a)** along with their average diurnal cycles **(b)** and the respective hourly average contributions **(c)**.

confirmed by high concentrations associated with increased wind speeds (Fig. S20), especially those that originate from the northern sector. These results (presented in the Fig. S20 for the full dataset) are in contrast to HOA which displays a much less diffuse spread, due to the intensity of local emissions (mainly traffic in the center of Athens). The distant signal of LV-OOA in the southeast direction could possibly be associated with processed aerosol derived from shipping activity (Petit et al., 2014) in the Aegean Sea.

Primary fossil fuel emissions (HOA) are very low during summer exhibiting a 5-fold decrease compared to the cold season; this is due to the fact that in July and August most of Athenians leave for their summer vacations, which reduces local traffic. Concentrations peak at around 07:00 and after 19:00 LT which corresponds to the early morning

and evening rush hours in downtown Athens. COA exhibits a slight hump during the lunchtime hours (13:00–15:00 LT) when concentrations rise to 65 % of the daily COA average after the morning minimum of around 50 % (also seen in the relative contribution of the factor), while a large nighttime peak is present at around 22:00 LT. This late peak, which is three times higher than the daily average value, is consistent with the late dinner hours and the operation of grill houses and restaurants in central Athens. SV-OOA exhibits 40 % higher concentrations during nighttime compared with the SV-OOA daily average, which apart from boundary layer dynamics may also be attributed to the condensation of semi-volatile compounds, as also implied by the excellent correlation of the factor with nitrate. During daytime, following the sharp decrease from the nighttime maxima, concentrations

remain close to 80 % of the daily average for some hours (10:00 to 14:00) before declining further in the afternoon. Finally, LV-OOA exhibits a peak during midday that is consistent with increased photochemical processes during the peak of solar radiation intensity (Fig. S2) which lead to further OA oxidation.

In summary, during the warm period, the vast majority (more than 80 %) of OA in the area is linked to SOA formation. The semi-volatile product is of mixed origin and is linked to quick atmospheric processes (within a few hours), such as photochemistry of primary sources, like biogenic emissions from vegetation, traffic emissions, or probably to a lesser extent regional biomass burning. This last assumption could be supported by the fact that OOA linked to aged BBOA has been reported at regional background sites in Greece (Bougiatioti et al., 2014) and elsewhere (Minguillon et al., 2015 TS9), as well as by the fact that during the warm season, air masses which mostly originate from the north or northeastern CE9 sector, carry pollutants from the Balkans and around the Black Sea, which are areas that are heavily impacted by wildfires from July to September (Sciare et al., 2008; Fig. S21). On the contrary, the low-volatility product is the result of more extensive oxidation of OA in the area, within a few days, and thus exhibits a more regional character.

*The cold period*: in this period, the selected solution stems from a three factor constrained run (HOA using $\alpha = 0.1$, COA using $\alpha = 0.2$, and BBOA using $\alpha = 0.4$) and consists of five factors: BBOA, HOA, COA, SV-OOA, and LV-OOA. The solution for winter 2016–2017 is presented (Fig. 6), while the respective solutions for winter 2013–2014 and 2015–2016 are provided in the Supplement (Fig. S13). The time series of the five PMF factors for winter 2016–2017 are shown in Fig. 6 along with their diurnal variability and the hourly contribution of each factor.

In terms of its affinity with RFPs found in the literature, HOA for the cold season in this study is found to exhibit excellent correlations with spectra obtained during the same season in earlier studies in Athens as well as other Greek cities (e.g., Patras; Florou et al., 2017) and also with HOA factors obtained in different environments, a fact also observed for the spectrum obtained in the warm season (Fig. S15). COA is excellently correlated with COA from Florou et al. (2017) in both Athens and Patras as well as with COA measured by Kaltsonoudis et al. (2017) (Fig. S16). When calculating the elemental ratios based on the study of Canagaratna et al. (2015) the O : C ratio for COA is 0.18, which is in accordance with the value of 0.11 derived for COA at the same site by Florou et al. (2017). BBOA exhibits a high correlation with factors from Zurich, Paris, and Finokalia as summarized in Fig. S17, while an excellent correlation is found when it is compared to BBOA found in Bologna, and earlier studies in Athens and Patras (Gilardoni et al., 2016; Florou et al., 2017). Once more, the calculated O : C ratio for BBOA is 0.25, which is in accordance with the

value of 0.27 derived for BBOA at the same site by Florou et al. (2017). The SV-OOA spectrum exhibits a high correlation with the average SV-OOA from Ng et al. (2011b), as well as with the IEPOX-OA from Budisulistiorini et al. (2013) ($r^2 = 0.80$ in both cases), as isoprene's main oxidation products such as methyl vinyl ketone and methacrolein are often used as biomass burning tracers (Santos et al., 2018). A similar correlation is also found with IEPOX-OA and SV-OOA during the winter 2015–2016 campaign. The factor exhibits a high correlation with SV-OOA from wintertime in Paris (Crippa et al., 2013) and SV-OOA from Hyytiälä (Äijälä et al., 2017) (Fig. S18). Finally, LV-OOA records an excellent correlation with the LV-OOA from Crippa et al. (2014), the average LV-OOA from Ng et al. (2011b), LV-OOA from Zurich during winter (Lanz et al., 2008), and with the oxidized OOA found in the region (Finokalia) (Bougiatioti et al., 2014; Fig. S19).

The identification of BBOA is mainly based on the two fragments of $m/z$ 60 and 73, considered as the "fingerprint" fragments of levoglucosan and biomass burning tracers. Indeed, BBOA exhibits an excellent correlation with these two fragments ($r^2 = 0.94$ and 0.9, respectively). Nss-K$^+$ is also proposed as a very good tracer for biomass burning and, as reported by Fourtziou et al. (2017), it shows a significant correlation with BC from wood burning (BC$_{wb}$), during wintertime in Athens. Consequently, the time series of nss-K$^+$ provided by the PILS-IC and $m/z$ 60 are studied together. It appears that during both winters (2013–2014 and 2016–2017) for which nss-K$^+$ data is available, $m/z$ 60 is in very good agreement with nss-K$^+$ ($r^2 = 0.85$; Fig. 7a). Furthermore, BBOA is highly correlated with BC$_{wb}$ ($r^2 = 0.77$), and exhibits a good correlation with nss-K$^+$($r^2 = 0.55$) and CO ($r^2 = 0.51$). SV-OOA correlates excellently with both wood burning "fingerprint" fragments of $m/z$ 60 and 73 ($r^2 = 0.99$ for both), highly with BC$_{wb}$ ($r^2 = 0.90$) and CO ($r^2 = 0.73$) (Fig. 7b), and it exhibits a good correlation with nss-K$^+$ ($r^2 = 0.55$); this demonstrates the direct link between SV-OOA and primary combustion sources (mainly biomass burning; Table S2). It can be seen in Fig. S21 that increased concentrations of both BBOA and SV-OOA are linked to air masses originating from northern and eastern Europe. During wintertime, these flow categories are associated with the prevalence of synoptic-scale northern winds and a decline in temperature in the area, leading to the appearance of PM episodes due to local combustion for residential heating (Paschalidou et al., 2015). The input of local sources confined in the Athens basin and in the vicinity of the sampling site is indicated by results of the wind analysis presented in Fig. S20. Markedly enhanced levels are associated with weak or stagnant conditions. These results are in contrast with those of Grivas et al. (2018) from a moderately populated area in the eastern part of the basin. They found that local biomass burning emissions played a less important role than advections from the northern part of the area. In the present case, in the densely populated center of Athens this

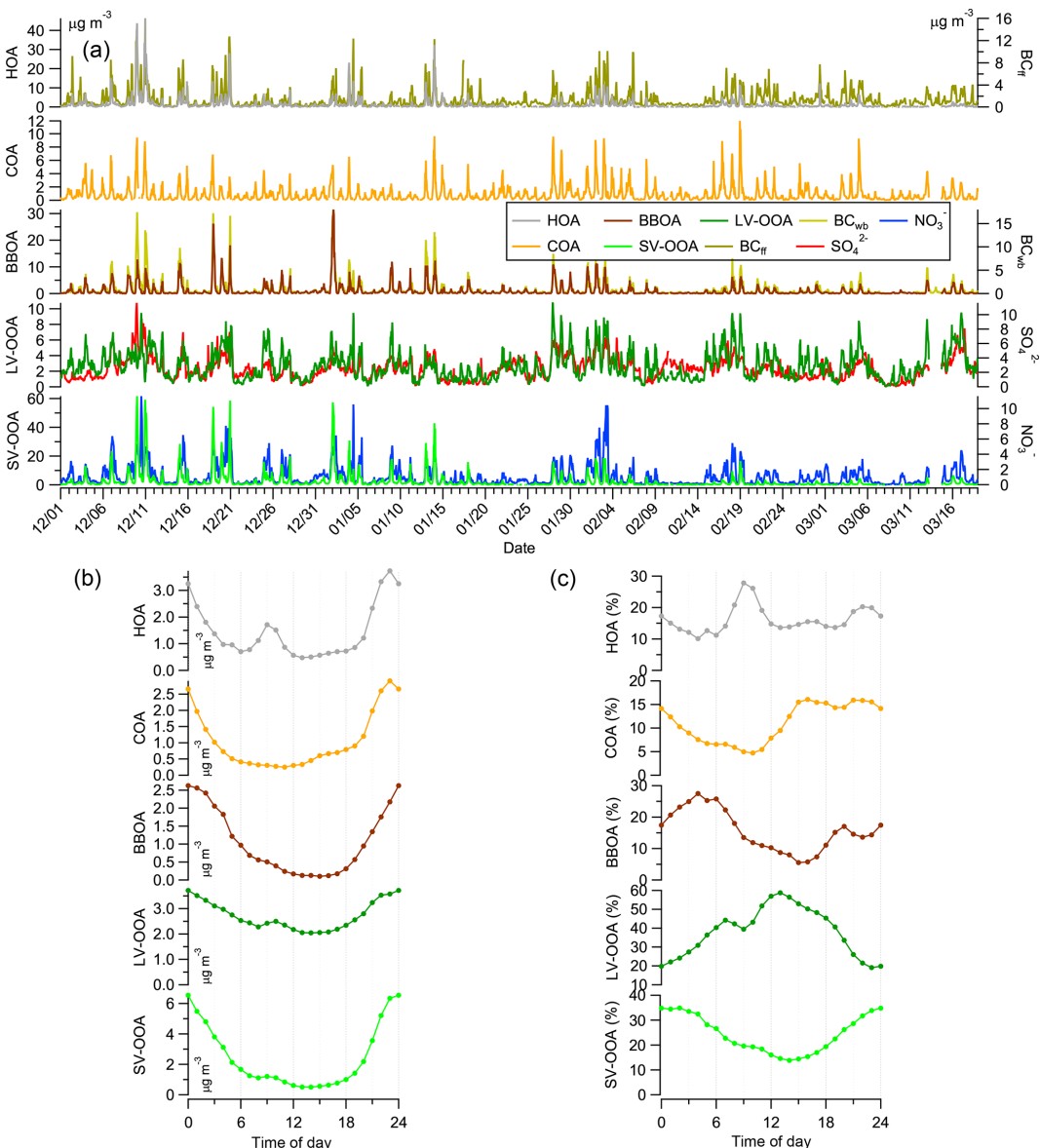

**Figure 6.** Time series of the contribution of the different factors identified by PMF between 21 November 2016 and 1 March 2017 **(a)** along with their average diurnal cycle **(b)** and respective hourly contribution **(c)**.

effect is less apparent. The local character of wood burning aerosols in dense residential areas in Athens has also been indicated by Argyropoulos et al. (2017).

Comparison of the HOA time series with BC and CO yields good correlations ($r^2 = 0.65$ and $r^2 = 0.65$, respectively). The factor is consistently more well correlated with $BC_{ff}$ than with $BC_{wb}$ (e.g., for the 2016–2017 $r^2$ is 0.60 vs. 0.52, respectively). Correlation of COA with nss-$K^+$ and chloride ($0.3 < r^2 < 0.4$) could indicate a minor influence from emissions derived from biomass burning in meat-cooking (Akagi et al., 2011; Kaltsonoudis et al., 2017). Finally, LV-OOA showed a good correlation with ammonium ($r^2 = 0.58$), nitrate ($r^2 = 0.61$), nss-$K^+$ ($r^2 = 0.4$), and

$m/z$ 73 ($r^2 = 0.51$), demonstrating that part of the very oxidized OA during wintertime may also originate from combustion sources.

Therefore, during the cold period, the OA in the area linked to SOA formation contributes around 65 % of the total organic fraction. In contrast to summer, the semi-volatile products seem to be linked to the fast oxidation of primary combustion emissions (e.g., BBOA), which is also reflected on its diurnal variability (Fig. 6) and in the strong correlations with external tracers of primary combustion (see Table S2). Affinity with biomass burning tracers highlights that the largest part of SV-OOA originates from the fast oxidation of BBOA. The low-volatility product is, in this case, likely of

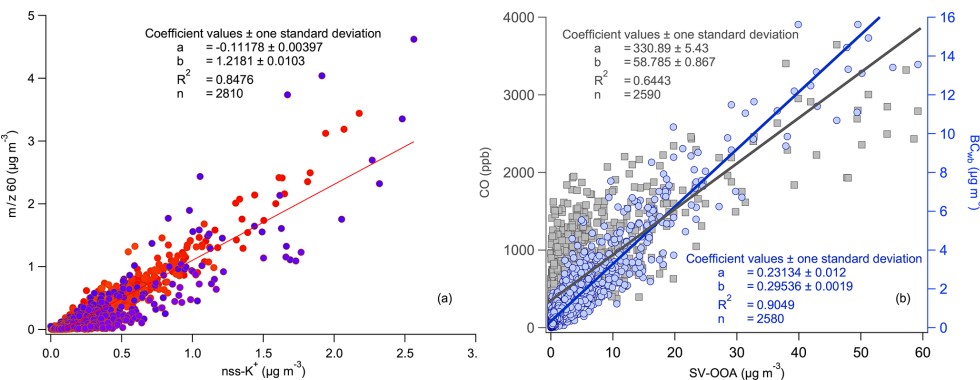

**Figure 7. (a)** The correlation of $m/z$ 60 with nss-K$^+$ for 2013–2014 (red) and 2016–2017 (blue), and **(b)** the correlation of SV-OOA with CO (grey) and BC (blue) for 2016–2017.

a more local than long-range transport nature, as also emphasized by the almost 2-fold higher values during nighttime.

The diurnal cycles of the five factors are shown in Fig. 6. HOA, originating from fossil fuel combustion, exhibits maximum values during nighttime, associated with combustion from central heating, and presents a secondary peak at 09:00 coinciding with the early morning traffic rush hour. The association of the factor with local primary emissions is also corroborated by the wind analysis plots (Fig. S20). The dependence of HOA on wind speed and direction is similar between cold and warm seasons. The concentration vs. wind speed distribution displays a wind dilution effect and is characteristic for traffic-related fine particles in Athens (Chaloulakou et al., 2003; Kassomenos et al., 2012).

COA has similar winter and summer diurnal profiles that display a moderate hump, with concentrations rising from 30 % (0.3 µg m$^{-3}$) to 60 % (0.6 µg m$^{-3}$) of the daily average (0.98 µg m$^{-3}$) during the lunchtime hours (12:00–15:00 LT) and a large nighttime peak (at approx. 22:00 LT); this is partly controlled by the decrease of the planetary boundary layer, but is also owing to the expected increase in the activity of numerous restaurants in the area. A similar diurnal cycle for COA was reported by Florou et al. (2017). BBOA is characterized by a pronounced diurnal cycle with peaking values during nighttime, associated with the production of this component in the evening by combustion for heating purposes. SV-OOA exhibits the largest diurnal amplitude, with nighttime values that are almost 6-fold higher compared to daytime. A plateau, with concentrations of SV-OOA around 50 % of the daily average value, following the sharp decline after midnight, is observed during the morning traffic rush hour, before another decline occurs until the daily minimum is reached at 14:00; this demonstrates the possibility of the factor's provenance from the oxidation of freshly-emitted primary combustion OA. Finally, LV-OOA also exhibits 2-fold higher values during nighttime compared with daytime. It has similar behavior to the SV-OOA factor, with a secondary peak at 10:00, exhibiting a 1 h lag after the morning

traffic rush hour; this once again shows that part of the low volatility OA may also originate from the fast oxidation of primary combustion emissions, as also implied by its correlation with combustion tracers.

Table 2 sums up the contribution of each one of the five identified factors during the three winters studied. Overall, during wintertime BBOA constitutes around 10 % of the total organic fraction. Based on the diurnal variability of this factor, its contribution is more pronounced during nighttime, when concentrations are 4-fold or higher than the daytime values, matching emissions from fossil fuel combustion represented by the HOA factor incorporating both traffic and heating oil combustion. Even though an exact mechanism is yet to be established, our assumption that the larger part of the SV-OOA comes from the rapid oxidation of freshly emitted BBOA through processes which involve nitrate radicals and/or heterogeneous reactions, appears justified by the excellent correlations with biomass burning tracers as well as by considering similar assessments found in other studies (Lathem et al., 2013; Cubison et al., 2011; Bougiatioti et al., 2014). In this manner the overall contribution of biomass burning becomes even more significant. Given that SV-OOA contributes around 30 % to the organic mass, it is evident that during wintertime, biomass burning may contribute almost half of the total OA, with this contribution reaching a maximum during nighttime. More specifically, for BBOA the lowest contribution during daytime is 5.5 %, whilst it reaches a maximum of 27.5 % during nighttime (Fig. 6). The same applies to SV-OOA with daytime minimum contribution of 13.8 % and a nighttime maximum of 34.9 %. It is also very important to note that even though the winter and summer mass spectra of SV-OOA have some similarities ($r^2 = 0.83$), there are also differences, especially in the origin of this component: during winter the majority of SV-OOA is linked to the oxidation of primary combustion sources, while during summer the absence of a significant correlation with BC or nss-K$^+$ implies the presence of different sources, both anthropogenic (but not biomass burning) and possibly biogenic.

**Table 2.** Contribution of the five organic aerosol components to the total organic fraction during the three individual winter campaigns.

|        | Winter 2013–2014 (18 December 2013–21 February 2014) | Winter 2015–2016 (23 December 2015–17 February 2016) | Cold 2016–2017 (1 November 2016–18 March 2017) |
|--------|--------|--------|--------|
| BBOA   | 12.4 % | 8.9 %  | 11.9 % |
| HOA    | 12.2 % | 9.7 %  | 16.4 % |
| COA    | 10.4 % | 8.1 %  | 11.7 % |
| SV-OOA | 19.8 % | 17.7 % | 28 %   |
| LV-OOA | 45.2 % | 55.6 % | 32 %   |

## 4   Summary and conclusions

High temporal-resolution measurements were conducted for an entire year (plus two, 2-month, intensive measurement campaigns during wintertime) at an urban background site in Athens, using an ACSM, a PILS-IC system, and an Aethalometer, in addition to routine pollution measurements. During the 16-month measurement period, several pollution events with $PM_1$ concentrations reaching as high as $220\,\mu\mathrm{g\,m^{-3}}$ were recorded, all encountered during the night in wintertime. In these cases, organics contributed the largest fraction to the submicron particulate mass, with the overall contribution during wintertime reaching 50 %, followed by sulfate ($\sim 20\,\%$) and BC ($\sim 14\,\%$). On a typical winter day, organics, BC, and nitrate double their concentrations during nighttime. The increase of the first two can be attributed to emissions linked to domestic heating, while nitrate exhibits higher concentrations due to the combined effect of decreased temperature and aerosol acidity, favoring partitioning in the aerosol phase. During summer, organics, BC, and nitrate concentrations are significantly lower, while sulfate and ammonium levels are increased. Organics are once more the main aerosol constituent contributing 46 %, followed by sulfate (30.5 %), ammonium (8.3 %), and BC (8 %). On a typical summer day, ammonium and sulfate concentrations peak at about $14{:}00\,\mathrm{LT}$ $(\mathrm{UTC}+2)$, which is consistent with secondary aerosol formation.

Organics, nitrate, chloride, and BC exhibited a clear seasonal cycle with a maximum during winter and a minimum during summer. Sulfate and ammonium exhibited an opposite cycle as the result of enhanced photochemistry, limited precipitation, and higher regional transport.

Based on the source apportionment of the OA, four factors were identified during summer, namely HOA, COA, SV-OOA, and LV-OOA, and five factors during winter, the same as in summer with the addition of primary biomass burning emissions (BBOA). During summer, HOA made up 4.3 % of the total organic fraction, COA comprised around 10 %, and the rest was linked to secondary organics (SV-OOA and LV-OOA). HOA had peak values during the morning traffic rush hour, and COA mainly peaked during nighttime. SV-OOA exhibited 2-fold higher concentrations during nighttime while LV-OOA exhibited a peak during midday,

which is consistent with photochemical processes. The semi-volatile product was clearly of mixed origin, was linked to quick atmospheric processing (within a few hours), and was comprised of VOCs emitted from primary sources like vegetation, traffic, and to some limited extent to processed regional biomass burning CE10. The low-volatility product, in contrast, was the result of more excessive oxidation, in the order of several days, and thus has a more regional character.

Combining the results from the three different winter campaigns, HOA accounts for almost 13 % of the organic fraction, COA for around 10 %, BBOA for 10 %, SV-OOA for 22 %, and LV-OOA for 45 %. All constituents exhibit significantly higher concentrations during nighttime, with HOA also being linked to primary emissions by heating oil combustion from central heating units and presenting a secondary peak during the morning traffic rush hours. COA has a similar diurnal profile to that observed during summer. BBOA is also characterized by a pronounced diurnal cycle with peaking values during the night from combustion for heating. SV-OOA has almost 6-fold higher concentrations during nighttime, consistent with its link to the oxidation of primary combustion sources, while even LV-OOA exhibits almost 2-fold higher concentrations during nighttime. In contrast to summer, the semi-volatile product during winter has a very clear origin, which is linked to the fast oxidation of primary combustion sources (HOA and BBOA); BBOA is the major source, due to the affinity of SV-OOA with biomass burning tracers. Part of the LV-OOA could also originate from the extensive oxidation of the local primary combustion sources, which shows that LV-OOA is of more local than regional character during winter.

In summary, it is clear that OA constitutes a large fraction of submicron aerosol throughout the year in the urban environment of Athens. During wintertime, a large part of this OA, as high as 50 %, originates from combustion sources for heating purposes, such as biomass burning and diesel oil fueled central heating, causing significant air quality deterioration. The nighttime contribution of BBOA is 7-fold higher than that during the day, while the respective contribution of SV-OOA is increased by a factor of 2.6. Given that fine PM concentrations reach up to $220\,\mu\mathrm{g\,m^{-3}}$ during wintertime, the significance of the contribution of these sources to air quality

degradation becomes even more striking, demonstrating the necessity for strategic, long-term mitigation actions.

*Data availability.* All data related to the publication are available upon request from Nikolaos Mihalopoulos (nmihalo@noa.gr).

*Supplement.* The supplement related to this article is available online at: https://doi.org/10.5194/acp-19-1-2019-supplement.

*Author contributions.* NM, EG, and AB conceived and facilitated the study. IS, AB, DP, EL, PZ, and MT contributed to the measurements. IS, AB, GG, and NM performed the analysis and wrote the paper. All authors commented on the paper.

*Competing interests.* The authors declare that they have no conflict of interest.

*Special issue statement.* This article is part of the special issue "CHemistry and AeRosols Mediterranean EXperiments (ChArMEx) (ACP/AMT inter-journal SI)". It is not associated with a conference. TS10

*Acknowledgements.* Iasonas Stavroulas and Nikolaos Mihalopoulos acknowledge support from the State Scholarship Foundation ("IKY Fellowships of Excellence for Postgraduate Studies in Greece–Siemens Programme, 2016-2017"), in the framework of the Hellenic Republic–Siemens Settlement Agreement. The authors would also like to acknowledge support from Francesco Canonaco and Andre Prévôt from PSI, who developed SoFi and provided valuable input related to positive matrix factorization. This study contributes to ChArMEx work package 1 on emissions and sources.

Edited by: François Dulac
Reviewed by: five anonymous referees

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

## Remarks from the language copy-editor

CE1    Please confirm or offer an alternative.

CE2    Please confirm or offer an alternative.

CE3    Please note that constrains has been changed to constraints in two instances in this section.

CE4    The sentence has been changed; please check that the meaning of the sentence is intact.

CE5    Please check.

CE6    Which winters are you referring to here? Please define.

CE7    Please check.

CE8    Please confirm.

CE9    Please confirm this edit or offer an alternative.

CE10    The sentence has been changed; please check that the meaning of the sentence is intact.

## Remarks from the typesetter

TS1    Copernicus Publications collects the DOIs of datasets, videos, samples, model code, and other supplementary/underlying material or resources as well as additional outputs. These assets should be added to the reference list (author(s), title, DOI, and year) and properly cited in the article. If no DOI can be registered, assets can be linked through persistent URLs. This is not seen as best practice and the persistence of the URL must be secured.

TS2    The composition of Figs. 1–6 has been adjusted to our standards.

TS3    Please specify if this is 2011a, b, or both.

TS4    2018 not mentioned in the reference list.

TS5    Please send a new Supplement pdf with renamed Sections, Figures, and Tables (S1, S2, etc.). Thank you.

TS6    Not mentioned in the reference list, please add.

TS7    Please confirm.

TS8    Should this be 2008? If not, it is not mentioned in the reference list.

TS9    Should this be 2016? If not, it is not mentioned in the reference list.

TS10    Please confirm.

TS11    Please provide publisher location.

TS12    DOI is apparently not working, please check.

TS13    Not mentioned in the text.

TS14    Not mentioned in the text.

TS15    Please provide last access date (d/m/y).

TS16    Not mentioned in the text.

TS17    Please provide page range or article number.

TS18    Not mentioned in the text.

TS19    Please provide page range or article number.