# Peer review of "Sources and processes that control the submicron organic aerosol in an urban Mediterranean environment (Athens) using high temporal resolution chemical composition measurements."

_Atmospheric Chemistry and Physics, 2018_

## Referee Comment (RC1) · Anonymous Referee #1 · 13 Jun 2018

The manuscript of Stavroulas et al., presents and analyzes the organic aerosol sources over Athens (Greece) using long term observations (1 year) and 2 intensive campaigns measurements. The source analysis is based on the Aerosol Chemical Speciation Monitor (ACSM). The finding are very important for the region of Athens but also crucial for the atmospheric science community as they found that during the winter a significant fraction of the secondary organic aerosol is clearly connected to the biomass burning organic aerosol. The paper is very well written and organized. I definetely recommend publishing of this paper, after some some minor changes:

[Figure]

(1) The introduction contains information about Athens topography and biomass burning aerosol (BBOA). However, since in the paper are discussed additional sources found (e.g., LV-OOA, SV-OOA, HOA and COA) I suggest adding a paragraph giving some information about these sources and provide previous measurements for Athens. For example Kostenidou et al., 2015 have measured the OA sources for Athens suburban area during an intensive summer (2012) camping using HR-AMS data.

(2) Page 3, line 62: please add Florou et al. 2017 after Kalogridis et al. 2017 reference.

(3) I suggest replacing the sentence: "Respective measurements using high resolution techniques are scarce and limited in time (Florou et al., 2017)" with: "Florou et al., 2017 have measured the chemical composition and the OA sources during a wintertime intensive campaign in the center of Athens using an HR-AMS an however their data are limited in time."

(4) Page 4, line 119: Please provide the RIENH4 and RIESO4 with the corresponding standard deviations.

(5) Page 6, line 162: the right parenthesis should not be in bold form.

(6) Page 9, line 261: please add domestic or residential before heating.

(7) Pages 12-14, lines 353-425 (warm period section): It would be interesting to compare the HOA, COA, SV-OOA and LV-OOA mass spectra of the warm period in terms of angle theta with the corresponding spectra of Kostenidou et al. (2015) that they found for a suburban area of Athens during summer.

(8) Pages 14-15 lines 426-459 (cold period section): It would be nice to also compare the BBOA, HOA, COA and LV-OOA mass spectra from the winter period with the BBOA, HOA, COA and OOA mass spectra of Florou et al. 2017 that they measured during the winter at the same site. Florou at al. (2017) found two BBOA factors during the winter for a study in Patras (Greece); one of them (BBOA-II) was less oxygenated and its origin was not fully explained i.e., it could be due to the different types of fuel or

combustion or due to different degree of BBOA aging. What is the angle theta between the SV-OOA mass spectrum of this study (that is linked to aged BBOA) and the BBOA-II of Florou et al. (2017)?

(9) Page 17, line 496: there is a "t" alone in that sentence which it should be deleted.

(10) For the winter case you show in Table 2 that the BBOA mass fraction is around 8-10%. How about the absolute mass concentration? Did you see any correlations with temperature?

(11) I believe that a comparison with summer sulfate measurements from previous years should be made. It could be a paragraph before the conclusion part. How did the PM1 sulfate mass concentration and mass fraction change over the years? You could use the data of Kostenidou et al. (2015), for Athens during the summer and Bougiatioti et al., (2014) and Hildebrandt et al., (2010), which are for summer but for the Finokalia station (different location) given the fact that the sulfate concentration is similar in many locations above Greece during the summer (Tsiflikiotou, Master thesis). Using this trend, could you make any implications? For example how did the economical crisis in affect the air quality (less industry that produces SO2, which is converted to particulate sulfate)?

(12) Figure 5. Please improve the resolution of this figure. Top graph: The left y axis should say "mass concentration ($\mu$g m-3)" once. The right y axis should also say "mass concentration ($\mu$g m-3)" once and for each sub-axis just indicate the name of the species without $\mu$g m-3. Bottom graphs: again "mass concentration ($\mu$g m-3)" or "% mass concentration and for each sub-axis just indicate the mane of the species without $\mu$g m-3 or Contrib.(%). Please take care the numbers on the y axis, some fall on other and it is difficult to be read. Avoid gaps.

(13) Figure 6. The same as for Figure 5.

(14) Figure 8. You should consider using a lighter green for the LV-OOA in order to be

more distinguishable from the SV-OOA. May be use another color for the map behind (light blue)?

References:

Bougiatioti, A., Stavroulas, I., Kostenidou, E., Zarmpas, P., Theodosi, C., Kouvarakis, G., Canonaco, F., Prévôt, A. S. H., Nenes, A., Pandis, S. N., and Mihalopoulos, N.: Processing of biomass-burning aerosol in the eastern Mediterranean during summertime, Atmos. Chem. Phys., 14, 4793-4807, 2014.

Florou, K., Papanastasiou, D. K., Pikridas, M., Kaltsonoudis, C., Louvaris, E., Gkatzelis, G. I., Patoulias, D., Mihalopoulos, N., and Pandis, S. N.: The contribution of wood burning and other pollution sources to wintertime organic aerosol levels in two Greek cities, Atmos. Chem. Phys., 17, 3145-3163, doi: 10.5194/acp-17-3145-2017, 2017.

Hildebrandt, L., Engelhart, G. J., Mohr, C., Kostenidou, E., Lanz, V. A., Bougiatioti, A., DeCarlo, P. F., Prévôt, A. S. H., Baltensperger, U., Mihalopoulos, N., Donahue, N. M., and Pandis, S. N.: Aged organic aerosol in the Eastern Mediterranean: the Finokalia Aerosol Measurement Experiment – 2008, Atmos. Chem. Phys., 10, 4167–4186, 2010.

Kostenidou, E., Florou, K., Kaltsonoudis, C., Tsiflikiotou, M., Vratolis, S., Eleftheriadis, K., and Pandis, S. N.: Sources and chemical characterization of organic aerosol during the summer in the eastern Mediterranean, Atmos. Chem. Phys., 15, 11355-11371, doi:10.5194/acp-15-11355-2015, 2015.

Tsiflikiotou, M. Spatial distribution of summertime particulate matter and its composition in Greece (Master thesis) http://nemertes.lis.upatras.gr/jspui/bitstream/10889/8345/1/MS%20Thesis%20M.Tsiflikiotou.pdf

---

## Referee Comment (RC2) · Anonymous Referee #2 · 13 Jun 2018

This manuscript presents a long-term dataset of near real time chemical composition of submicron aerosols in Athens, Greece. It is completed by two intensive campaigns during winter time. Statistical analysis was performed in order to apportion the sources of organic matter. The subject of this paper is of interest and falls within the scope of ACP, although in its current form neither the methodology (PMF) nor the results bring strikingly new outputs in this region. I am still favorable to publication after major revisions.

**Major Comments** 1) Overall, the result section is too descriptive. Describing the

angles between profiles impinges upon the actual results. The authors should re-focus the discussions on how this study slots into previous knowledge in Greece (Athens and their cities, such as Patras) and, why not, in the Eastern Mediterranean. Moreover, strong assessments are made regarding SOA, simply from diurnal variations. The authors should either tone down these statements or add much more discussion and figures. Then, organonitrates have been found to have significant contributions in Greece (Florou et al., 2017). Can the authors add some more knowledge about that? (especially regarding the role of primary combustion sources in their formation?) 2) The method used by the authors to select the appropriate solution is not clearly stated, although it was inspired by Crippa et al. (2014). From the first ACSM intercomparison, Frohlich et al. (2015) proposed a methodology of find optimal solution, and stated some recommendations. Why did the authors prefer Crippa et al.? 3) The local vs regional vs advected features are not well characterized, and I would strongly recommend the authors to perform a wind analysis, especially for the "local" sources (eg traffic & biomass burning).

**Minor comments** - P3 l78: the introduction mainly focuses on wintertime biomass burning, so why would you need long-term datasets? - P4 l120: please indicate the calibration values - P4 l103: the ACSM does not measure "aerosol mass" but only the chemical composition; it is not equivalent to a TEOM-FDMS. - P5 l123: it is not clear why a chemical-dependent CE has not been applied. Although it is discussed later on, it could be quickly stated here. - P5 l125: did the authors use denuders ahead of the PILS in order to prevent nitric acid, sulfuric acid and ammonia to be respectively confused with particulate nitrate, sulfate and ammonium? - P5: filter samplings are not presented. - P10 l292: See major comment. I don't think that the only fact that nitrate has a morning peak similar to BC is enough to link it with morning traffic. More discussion would be needed. - P11 l318-319: Datasets have been separated into cold and warm months prior to PMF. Why not seasonally? Not just cold and warm months influence the characteristics of secondary organic aerosols, it could also be related to air masses. So the approach chosen here is not well justified. - P11 l329:

why the HOA from Ng et al. has been used? Why not any other profiles, especially gotten from previous studies in Greece? - P11 l333: Do the authors have any hint of how representative the BBOA of Ng et al. is in Greece? - P12 l340-341: it could be appreciable if the authors provide a bit more details on the metrics used through the correlation of PMF timeseries with external tracers. - P12 l351: a correlation coefficient of 0.86 corresponds to a $r^2$ of 0.63, which is still a good statistical correlation. Ranges of $r^2$ are by the way not consistent throughout the manuscript. P13 l397 and p14 l400, $r^2$ of 0.32, 0.36 and 0.39 are considered as "moderate", which should rather be a poor correlation. Later, p16 l477, a $r^2$ of 0.53 is considered as "very well". I strongly suggests the authors to use a consistent description of correlation coefficients. - P13 l388-390: See major comment. linking SVOOA with primary sources only from mean diurnal variations is not convincing. Please add more discussion. - P14 l400: See major comment. Same comment, the statement "SVOOA may, to some extent, partially originate from a combustion source" seems random and hardly quantifiable. - P14 l407: "HOA emissions are very low", compared to what? - P14 l422: the authors would need to prove the link between SOA and regional biomass burning. - P15 l444-448: how does BBOA compare with BBOA profiles from other studies in Greece? Or in other Mediterranean sites? - P16 l477-479: HOA correlates moderately with CO, BC and NO. So is HOA representative of traffic?

**Technical corrections and suggestions** - P1 l22: replace "fine" by "submicron" - P1 l24: rephrase to "with concentrations during wintertime sporadically reaching up to 200 $\mu$g/m3". Please also indicate the time resolution for this (daily/hourly concentrations?) - P2 l50: replace "namely" by "such as" - P4 l92: "105 m above sea level" - P4 l105-109: these information are redundant and/or well known. It could be removed. - P6 l165-172: I don't think a thorough description of PMF and ME-2 is necessary here. Please shorten or remove this section. - P8 l237: rephrase to "The other striking feature is that" - P8 l241: rephrase to "average 8 of such" - P8 l242: please add "to our knowledge" - P8 l243: rephrase to "highlight the strong impact" - P10 l286: rephrase to "to the regional character" - P11 l231: one could cite here Canonaco et al.(2015): Canonaco,

F., Slowik, J. G., Baltensperger, U., and Prévôt, A. S. H.: Seasonal differences in oxygenated organic aerosol composition: implications for emissions sources and factor analysis, Atmos. Chem. Phys., 15, 6993-7002, https://doi.org/10.5194/acp-15-6993-2015, 2015. - P12 l343-350: I think this has already been presented elsewhere, so I don't think it is necessary here.

---

## Referee Comment (RC3) · Anonymous Referee #3 · 19 Jun 2018

The manuscript "Sources and processes that control the submicron organic aerosol in an urban Mediterranean environment (Athens) using high temporal resolution chemical composition measurements" presents the submicron aerosol chemical composition in Athens, Greece. In addition to study the seasonal variation of the main chemical species, organics, sulfate, nitrate, ammonium, chloride and black carbon, the sources of organics were assessed by statistical methods using Positive Matrix Factorization (PMF). The results of PMF showed that in winter there were five factors for organic aerosol (OA); fossil fuel combustion (HOA), biomass burning (BBOA), cooking (COA)

and two different oxidized organic aerosols (SV-OOA and LV-OOA), of which primary sources were pronounced. In summertime, most of the OA was associated with oxidized factors representing secondary organic aerosol.

This paper exploits an extensive data set (more than a year of data) and the instruments used are present-day. However, the results of this study follow very closely to those presented previously for urban areas in winter and summer not revealing any novel sources of aerosols or phenomena in urban area. My main concern is though the PMF/ME2 analysis. Authors found biomass burning and cooking factor by constraining them with reference mass spectra. My feeling is that any factor can be constrained and a mass fraction of $\leq$10% is obtained for that factor even though there is no clear evidence of the existence of that factor. A standardized methodology to perform source apportionment on AMS data using the ME2 is given in Grippa at al. (2014) but since the authors do not show the results (residues) without constraining factors, or constraining only HOA, I can't be sure that the given methodology has been followed. My fear is that authors discovered factors that do not exist (especially COA). As it is discussed Mohr et al. (2012) the actual differentiation between AMS aerosol spectra from cooking and traffic (or BBOA) is difficult for unit mass resolution spectra (ACSM data), and it is mostly based on the relative abundances of signals at m/z 55 and 57. Authors need to provide the evidence of COA more carefully. According to Crippa et al. (2014) the presence of the meal hour peaks is necessary to support COA at least in urban areas. In the paper of Stavroulas et al. it is stated that COA exhibits a slight hump during lunchtime but this hump is very difficult to see from the figures. COA as well as all the other PMF factors, except LV-OOA, had largest concentrations in nighttime. If meteorology (boundary layer height) affects that much on concentrations, PMF analysis can be very tricky and it may not be possible to distinguish all the sources, and that needs to be acknowledged in the paper.

I think that the data presented in this paper in worth publishing. However, major changes need to be done before this paper merits publication in ACP. I recommend

that authors redo PMF analysis according to Crippa et al. (2014) and consider the validity of BBOA and COA in every step (and show results from every step in supplement). Additionally, I suggest authors to concentrate on novel results that interest the whole scientific community not just Athens area, and state it clearly what are the new findings presented in this paper.

Major comments

1. Page 2-3, Introduction; Introduction section concentrates too much on Athens area and do not give general introduction to the research questions and issues related. I suggest taking more global point of view to the topic in introduction.

2. Page 11; "3.3. Source apportionment of organic aerosol" section is too long. Because the methods (PMF/ME2) are quite commonly used nowadays, and described in the literature, this section needs to be shorten or moved to experimental or supplement leaving only clear results to "Results and Discussion" section. Authors used ME2 traditional way so there are no scientifically new results in this section regarding the use of ME2.

3. Page 11, line 323; unconstrained runs, the results from unconstrained runs need to be presented in supplement. It is very difficult for the reader to trust the results (especially BBOA and COA factors) if unconstrained results are not shown. The technical guidelines for constraining are given in Crippa et al. (2014) and the results for each step needs should be presented.

4. Page 12; affinity between spectra by ïĄśïĂăangle approach, why did you use this approach here and Pearson correlation (with R2 earlier)? It is very confusing for readers that are not familiar with this angle approach. I suggest to use Pearson correlations (R2) throughout the manuscript.

5. Meteorological parameters; meteorological parameters are not given in the paper. Please provide at least temperature, radiation and boundary layer height that are im-

portant regarding the concentrations and the sources of aerosol

Minor comments

6. Page 1-2, Abstract; line 30-31; "These results highlight the rising importance of biomass burning in urban environments during wintertime." The contribution of biomass burning to organics was 10% in wintertime. It's quite a small contribution. This sentence needs an evidence or to be modified.

7. Page 3, line 82; "non-refractory part"; you also measured BC, why it is not included in main objectives (BC is refractory component)?

8. Page 4, line 101-102; "s/n 140-139" not needed here

9. Page 4, line 102; Aerodyne Research Inc.

10. Page 4, lines 112-120; "The instrument has participated in an intercomparison study..." This information is not relevant. Please remove this intercomparison section or move it to supplement.

11. Page 4, line 118-120; give RIE values

12. Page 5, line 122-123; default collection efficiency of O.5, please use equation of Middlebrook et al., (2012) to calculate composition dependent collection efficiency.

13. Page 5, line 138-139; more information is needed on SMPS measurements; size range, how number size distribution was converted to mass concentration (density)?

14. Page 5, line 140-144; give more details of selected absorption exponents, are they default values or did you calculate them specifically from this data set/ for this location?

15. Page 5, line 144; remove "Necessary"

16. Page 5, line 145; remove "historic"

17. Page 6, line 160; on the organic mass spectra obtained

18. Page 7, line 185; "following section"; give the number of sections

19. Page 7, line 194-196; describe PM2.5 filter collection and thermal-optical method in experimental section

20. Page 8, line 223; add time base for averages e.g. 1-hour average

21. Page 8, line 244, change "to the levels" to "on the levels"

22. Page 8-9, line 244-247; "These observations are in accordance..." this sentence is unclear and needs to be modified

23. Page 9, line 261-262; "additional primary emissions from heating play a role", based on what? Explain how you see this addition in results.

24. Page 9, line 273; what are increased local sources for nitrate in winter?

25. Page 11, line 309-312; "higher organics concentration during early night could possibly be due to biogenic/vegetation sources that produce volatile components that condenses on particulate phase during night." This assumption needs evidence, maybe reference or can you see this in mass spectra of organics?

26. Page 12, line 354-356; if HOA; COA; SV-OOA and LV-OOA are mentioned here for the first time the long names should be given. Please double-check when abbreviations are given for the first time.

27. Page 13, line 383-385; "OA precursors are maximum during night similar to SV-OOA". Please give reference or results.

28. Page 13, line 385-387; "SV-OOA shares some similarities with SOA from diesel exhaust". This is too vague. Give correlation coefficient or remove sentence. How much diesel vehicles there are in Athens?

29. Page 13, line 3963-397; "COA shown moderate correlation with nitrate". Explain why.

30. Page 14, line 403; Is figure number here really 8? Double-check figure numbers.

31. Page 14, line 410; "COA exhibits a slight hump during lunch hours." I really can't see this hump in Figure 5. There is similar lump between 4 and 9 am. How do you explain this morning lump? Please add negative standard deviations to Figure 5 (and all the other figures as well) because it's confusing (and maybe misleading) when only positive deviations are shown. Add also zero-lines to Figure 5 and Figure 6.

32. Page 14, line 417; "moderate hump for SV-OOA during mid-day". I can't see this hump in Figure 5. If you think this "hump" is true show it with numbers e.g. how much SV-OOA increased during mid-day compared to e.g. morning.

33. Page 16, line 463; How did you calculate Nss-K?

34. Page 16, line 467-471; "SV-OOA mass spectra includes also fingerprint fragments of biomass burning m/z 60 and 73"; what fraction of these mass fragments were associate with BBOA and SV-OOA (and other factors)?

35. Page 16, line 477-478; why COA correlates with potassium and chloride?

36. Page 16, line 484-490; "SV-OOA in cold period is linked to the fast oxidation of primary combustion sources (BBOA and HOA) which is also reflected on its diurnal variability." This sentence needs explanation and proof.

37. Page 17, line 494-495; "moderately hump for COA during lunchtime". This cannot be seen in Figure 6.

38. Page 17, line 499-500; "A moderate peak during the morning traffic hour (partly masked by the high night values) for SV-OOA," This peak is very difficult to see in Figure 6 (concentrations) and it does not exist in contributions figure. Please, re-consider how you define peaks/humps etc. in the paper.

39. Page 17, line 510-513, "SV-OOA comes from the rapid oxidation of freshly emitted BBOA", this needs more explanation. What is the oxidation process, what are the

oxidants in wintertime? In general, it said that SV-OOA is linked to quick atmospheric processing of VOCs within few hours. This needs to be explained in more detail (with results).

40. Page 18, line 533-534; "organics, BC and nitrate double their concentrations during night-time as a results of additional primary combustion for heating purposes." Do you suggest that nitrate and BC are mostly from heating? I think that the increase in winter in nighttime is mostly due to boundary layer change.

41. Page 19, line 557-559; "HOA being affected by combustion from central heating", The impact of central heating was not discussed in Results section. If the authors think that this is the source of HOA it should be discussed and (justified) earlier.

42. Figure 1; Add "1-hour averaged" mass concentrations

43. Figure 4; in upper figure you use "organic aerosol" but in lower figure "Organics". Please be consistent with the names.

44. Figure 6; why did you plot COA and nss-K to the same figure? Based on the time series they correlate quite well. Do you suggest that they originate from the same source?

45. Table 1; please give the name of the month clearer way e.g. using Jan, Feb etc.

Technical comments:

46. Page 6, line 163; time series

References

Crippa, M., Canonaco, F., Lanz, V. A., Äijälä, M., Allan, et al. Organic aerosol components derived from 25 AMS datasets across Europe using a newly developed ME-2 based source apportionment strategy. Atmos. Chem. Phys. 14, 6159–6176, 2014.

Middlebrook, A. M., Bahreini, R., Jimenez, J. L., and Canagaratna, M. R.

Evaluation of composition dependent collection efficiencies for the aerodyne aerosol mass spectrometer using field data, Aerosol Sci. Tech., 46, 258–271, https://doi.org/10.1080/02786826.2011.620041, 2012

Mohr, C., DeCarlo, P.F., Heringa, M. F., Chirico, R., Slowik, J. G. et al. Identification and quantification of organic aerosol from cooking and other sources in Barcelona using aerosol mass spectrometer data. Atmos. Chem. Phys., 12, 1649–1665, 2012.
* * *

---

## Referee Comment (RC4) · Anonymous Referee #4 · 22 Jun 2018

The manuscript presents a one-year dataset (2016/2017) of near real time chemical composition of submicron aerosol particles measured in Athens and its subsequent PMF analysis. This dataset is complemented by 2 intensive campaigns carried out in winter (2013/2014 and 2015/2016). While these data are of prime interest, the manuscript is very descriptive and do not bring significant new results for the scientific community. However, I support the publication of this manuscript after major modifications.

1/ The PMF analysis and the constrains applied are somewhat confusing and the

methodology should be described more clearly and in a more systematical way. A lot of different alpha values are selected (arbitrarily?) for the different factors. For a given source profile authors choose different alpha values for the different dataset. This must be explained and justified. Did the authors studied the influence of the alpha values on the sources contributions in a more systematic way? An alpha value of 0.1 is, from my point of view, too low for COA. Same for HOA, an alpha value of 0.05 is, in a first approach, too low considering the variability of the vehicular fleet (diesel/gasoline share, ...).

2/ Authors should convince the reader of the validity of the COA factor extracted from their analysis. The COA factor extracted here from the PMF analysis represents a contribution as high as BBOA in winter. It seems well correlated with the BBOA factor and other combustion markers (nssK+ for instance) and do not exhibit the classical midday hump. As the COA MS profile contains a slight contribution of m/z 60, I suspect a mix of both COA and BBOA factor. Also, the reference mass spectra chosen to constrain COA has been obtained in Paris. In Paris, the main site was located in the local Chinatown and was surrounded by well-known fast food brands. One could assume that the cooking emissions in Athens are slightly different than those of Paris for this specific study.

3/ The split of the data series between warm and cold period sounds quite arbitrary. Does it actually rely on temperature? If yes, this should be explicitly discussed in the text. While necessary for such long data series, splitting the dataset can induce a discontinuity of the sources contributions. Are such discontinuities observed here?

4/ If the data are available, I strongly suggest that the authors carry out a local winds analysis. From my experience such high nocturnal peaks are often mostly associated to local wind changes and in this case the occurrence of nocturnal breezes. In such cases (heavily polluted urban area), a local wind analysis is, from my point of view, much more relevant than a long-range transport analysis. Also, the influence of local wind patterns can induce strong correlation within the dataset which can not be related

to sources intensities or atmospheric transformation processes.

---

## Referee Comment (RC5) · Anonymous Referee #5 · 28 Jun 2018

This paper aims to identify sources of submicron organic aerosols in Athens with a major interest on quantifying the contribution of biomass burning. Results are based on high temporal resolution chemical measurements performed by an ACSM. As stated by authors, this is the first study on submicron aerosol by using high temporal measurements during a relatively long period (1 yr plus 2 winter periods). However, it is a very descriptive work that does not provide new knowledge on atmospheric processes and sources in the eastern Mediterranean. The study is focused on the organics and mainly in the contribution of wood burning, as stated in the introduction section and as

deduced from the extension of measurements during the two winter periods. Impact of wood burning in air quality is a growing concern in Athens in the last years. Thus, the authors (5 of 7) co-authored a paper currently on ACPD (https://doi.org/10.5194/acp-2018-163) focused on the impact of residential heating on fine particulate matter by applying PMF to the chemical characterization of filters (24 and 12h resolution). This study was performed in the same place and during part of the period covered by the present study. I have read the comments by the other reviewers and I strongly agree with the remarks form RC2, and also 3 and 4.I would like to add some minor comments and insist on some of the comments already mentioned by the other referees. My major concern is the use of constrains based on measurements performed in very different areas (HOA, COA and BBOA form north Europe) for the PMF of organics. Are these profiles usable in the study area? The profiles used should be more similar to the profile emissions in the area. Do the authors have some information about COA and BBOA profiles from the eastern Mediterranean area? Most statements about the origin the origin of the SVOOA and SOA are hard to demonstrate based only in the interpretation of the diurnal variation.

Minor comments Experimental methods; Page 5. Was the ACSM calibrated on field? No information about filters sampling and analysis is provided. Please, indicate sampling period and frequency and the methods of filters treatment and analysis. Please, indicate the size range of SMPS TSI3034. Results and discussion In the supplementary, authors show the correlations between filters and ACSM for the whole period (SL1.1) and for the winter periods (SL1.2). Is there any reason for the different slopes determined for each period? Do you expect the presence of coarse nitrate in the 1-2.5 um fraction? Did the OA/OC ratio keep constant along the sampling period? Did you compare EC vs BC? Is this ratio constant along the study period? Is any difference in winter with respect summer? Line 256. Contribution of nitrate in summer? Line 260. There is a BC peak in June not related to any other compound (figure 3). What is the cause of this maximum? Any information from the measurements by means of Aethalometer? Line 265 semi-volatile inorganics; and organics? Lines 290 293: Is

nitrate primary emitted? Do you mean that nitrate is quickly formed from primary NOx? Can be the relatively high levels of nitrate be related to the low stability of nitrate with temperature? It is risky to assign a source origin to nitrate only form the diurnal variation. Line 296: What do you mean with "normalizing the diurnals? Line 363: Please, replace "2016 and 17" by "2016 and 2017" Line 477. Did you check the correlation with the BC factions? Does the HOA factor correlated better with BCff than with BCwb? Figure 8. Why COA factor increased with wind form the eastern sector? Line 480-483. During the cold period nitrate correlates with LV-OOA while in summer it correlated with SV-OOA. Could you explain the reasons of it? Summary and conclusions Line 535. Sulfate and ammonium concentrations are not lower in summer Line 571. What do you mean with "central heating"? Fuel-oil heating?

---

## Author Comment (AC1) · 23 Aug 2018

Atmos. Chem. Phys. Discuss., https://doi.org/10.5194/acp-2018-356-RC1, 2018 © Author(s) 2018. This work is distributed under the Creative Commons Attribution 4.0 License.

Response to Anonymous Referee #1 comments

The manuscript of Stavroulas et al., presents and analyzes the organic aerosol sources over Athens (Greece) using long term observations (1 year) and 2 intensive campaigns

measurements. The source analysis is based on the Aerosol Chemical Speciation Monitor (ACSM). The finding are very important for the region of Athens but also crucial for the atmospheric science community as they found that during the winter a significant fraction of the secondary organic aerosol is clearly connected to the biomass burning organic aerosol. The paper is very well written and organized. I definetely recommend publishing of this paper, after some minor changes:

Response: We thank the anonymous referee for the thoughtful review. We have further elaborated on these points in the revised manuscript.

(1) The introduction contains information about Athens topography and biomass burning aerosol (BBOA). However, since in the paper are discussed additional sources found (e.g., LV-OOA, SV-OOA, HOA and COA) I suggest adding a paragraph giving some information about these sources and provide previous measurements for Athens. For example Kostenidou et al., 2015 have measured the OA sources for Athens suburban area during an intensive summer (2012) camping using HR-AMS data.

Response: Indeed, the introduction focuses mostly on winter time conditions and mainly discusses the biomass burning related atmospheric chemistry and its effect on air quality. The introduction in the revised version of the manuscript now adopts a wider perspective and addresses the referee's comment.

(2) Page 3, line 62: please add Florou et al. 2017 after Kalogridis et al. 2017 reference.

Response: Reference added.

(3) I suggest replacing the sentence: "Respective measurements using high resolution techniques are scarce and limited in time (Florou et al., 2017)" with: "Florou et al., 2017 have measured the chemical composition and the OA sources during a wintertime intensive campaign in the center of Athens using an HR-AMS an however their data are limited in time."

Response: Amended.

(4) Page 4, line 119: Please provide the RIENH4 and RIESO4 with the corresponding standard deviations.

Response: Response factor and relative ionization efficiencies of NH4+ and SO4= are now added to the revised supplementary material.

(5) Page 6, line 162: the right parenthesis should not be in bold form.

Response: Amended.

(6) Page 9, line 261: please add domestic or residential before heating.

Response: Amended.

(7) Pages 12-14, lines 353-425 (warm period section): It would be interesting to compare the HOA, COA, SV-OOA and LV-OOA mass spectra of the warm period in terms of angle theta with the corresponding spectra of Kostenidou et al. (2015) that they found for a suburban area of Athens during summer.

Response: A comparison with the summertime factors from Kostenidou et al. (2015) has now been added. Primary factors correlate well, namely HOA with R2=0.92 for the 2016 dataset and R2=0.94 for 2017, COA with R2=0.75 for 2016 and R2=0.77 for 2017. On the other hand, the semi-volatile component correlates moderately (R2=0.50 and 0.56 for 2016 and 2017 respectively), mainly because of the least oxidized nature of this study's SV-OOA which is probably related with the fact that the Thissio station is urban (city-center) in contrast to the suburban Demokritos station where the measurements of Kostenidou et al. (2015) took place. Finally, the LV-OOA factor shows a slightly better correlation (R2=0.61 and 0.59 for 2016 and 2017 respectively). The fact that the LV-OOA factor does not exhibit better correlation, is driven mostly by the elevated signal at m/z=18 (H2O+) attributed to this factor in this study, in contrast to its almost complete absence in the Kostenidou et al. (2015) spectra, performed with a High Resolution Time-of-Flight Aerosol Mass Spectrometer. If m/z=18 of this study is excluded from the correlation exercise the derived values for R2 are of 0.85 and 0.84 for 2016 and 2017

respectively.

(8) Pages 14-15 lines 426-459 (cold period section): It would be nice to also com-pare the BBOA, HOA, COA and LV-OOA mass spectra from the winter period with the BBOA, HOA, COA and OOA mass spectra of Florou et al. 2017 that they measured during the winter at the same site. Florou at al. (2017) found two BBOA factors during the winter for a study in Patras (Greece); one of them (BBOA-II) was less oxygenated and its origin was not fully explained i.e., it could be due to the different types of fuel orcombustion or due to different degree of BBOA aging. What is the angle theta be-tween the SV-OOA mass spectrum of this study (that is linked to aged BBOA) and the BBOA-II of Florou et al. (2017)?

Response: A comparison of the mass spectra for HOA, COA, BBOA and LV-OOA to the ones obtained by Florou et al (2017) for Athens has been added to the revised version. Correlations are very good for all primary factors, e.g. HOA with R2=0.92, COA with R2=0.96 and BBOA with R2=0.88. For the LV-OOA factor the same issue as when comparing with the summertime factors arises. When taking into account the m/z=18 fragment R2 is 0.57 while when excluding it correlation is stronger with R2=0.78. For the Patras site, as well, the correlation of the primary factors is very good (HOA: R2=0.97, COA: R2=0.93 and BBOA: R2=0.89). LV-OOA correlates well when not taking into account the contribution of m/z=18 (R2=0.88) while correlation less strong otherwise (R2=0.68)

(9) Page 17, line 496: there is a "t" alone in that sentence which it should be deleted.

Response: Amended.

(10) For the winter case you show in Table 2 that the BBOA mass fraction is around 8-10%. How about the absolute mass concentration? Did you see any correlations with temperature?

Response: As seen from the factor's time-series, BBOA exhibits the highest concentrations during nighttime, when temperatures are lower and people resort to biomass burning for heating purposes. However, temperature alone cannot explain the high values of BBOA as high values have been also observed during smog periods (SP, Fourtziou et al., 2017), characterized by low wind speed, which does not allow dispersion of air masses.

(11) I believe that a comparison with summer sulfate measurements from previous years should be made. It could be a paragraph before the conclusion part. How did the PM1 sulfate mass concentration and mass fraction change over the years? You could use the data of Kostenidou et al. (2015), for Athens during the summer and Bougiatioti et al., (2014) and Hildebrandt et al., (2010), which are for summer but for the Finokalia station (different location) given the fact that the sulfate concentration is similar in many locations above Greece during the summer (Tsiflikiotou, Master thesis). Using this trend, could you make any implications? For example how did the economical crisis in affect the air quality (less industry that produces SO2, which is converted to particulate sulfate)?

Response: We would like to thank the reviewer for his/her suggestion., The studies of Bougiatioti et al. (2014), Kostenidou et al. (2015) and this study were performed well within the economic recession, while only the short-term study of Hildebrandt et al. (2010) was performed before. . Thus by comparing levels at different locations, different months and given the limited amount of data such comparison could be biased. There is, indeed, an apparent reduction in the mean annual submicron sulfate levels, when compared with previous filter-based studies conducted in the area during the previous decade (Theodosi et al., 2011; Pateraki et al., 2012). We will keep however his/her suggestion and we plan in the future to collect all available data set from Greece to examine the impact of economic recession on SO4 levels.

(12) Figure 5. Please improve the resolution of this figure. Top graph: The left y axis should say "mass concentration (g m-3)" once. The right y axis should also say "mass concentration (g m-3)" once and for each sub-axis just indicate the name of the

species without g m-3. Bottom graphs: again "mass concentration (g m-3)" or "% mass concentration and for each sub-axis just indicate the mane of the species without g m-3 or Contrib.(%). Please take care the numbers on the y axis, some fall on other and it is difficult to be read. Avoid gaps.

Response: Figure 5 has now been redrawn in order to make the diurnal variability of each factor clearer. The referee's suggestions/comments have been taken into account.

(13) Figure 6. The same as for Figure 5.

Response: Figure 6 has been redrawn in the same manner as Figure 5.

(14) Figure 8. You should consider using a lighter green for the LV-OOA in order to be more distinguishable from the SV-OOA. May be use another color for the map behind (light blue)?

Response: Figure 8 has been redrawn in the revised manuscript, so the information depicted is more clear.

References

Bougiatioti, A., Stavroulas, I., Kostenidou, E., Zarmpas, P., Theodosi, C., Kouvarakis, G., Canonaco, F., Prévôt, A. S. H., Nenes, A., Pandis, S. N., and Mihalopoulos, N.: Processing of biomass-burning aerosol in the eastern Mediterranean during summertime, Atmos. Chem. Phys., 14, 4793-4807, https://doi.org/10.5194/acp-14-4793-2014, 2014.

Florou, K., Papanastasiou, D. K., Pikridas, M., Kaltsonoudis, C., Louvaris, E., Gkatzelis, G. I., Patoulias, D., Mihalopoulos, N., and Pandis, S. N.: The contribution of wood burning and other pollution sources to wintertime organic aerosol levels in two Greek cities, Atmos. Chem. Phys., 17, 3145-3163, https://doi.org/10.5194/acp-17-3145-2017, 2017.

Fourtziou, L., Liakakou, E., Stavroulas, I., Theodosi, C., Zarmpas, P., Psiloglou, B., Sciare, J., Maggos, T., Bairachtari, K., Bougiatioti, A. and Gerasopoulos, E., 2017. Multi-tracer approach to characterize domestic wood burning in Athens (Greece) during wintertime. Atmospheric Environment, 148, pp.89-101.

Hildebrandt, L., Kostenidou, E., Mihalopoulos, N., Worsnop, D.R., Donahue, N.M. and Pandis, S.N., 2010. Formation of highly oxygenated organic aerosol in the atmosphere: Insights from the Finokalia Aerosol Measurement Experiments. Geophysical Research Letters, 37(23).

Kostenidou, E., Florou, K., Kaltsonoudis, C., Tsiflikiotou, M., Vratolis, S., Eleftheriadis, K., and Pandis, S. N.: Sources and chemical characterization of organic aerosol during the summer in the eastern Mediterranean, Atmos. Chem. Phys., 15, 11355-11371, https://doi.org/10.5194/acp-15-11355-2015, 2015.

Pateraki, S., Assimakopoulos, V.D., Bougiatioti, A., Kouvarakis, G., Mihalopoulos, N. and Vasilakos, C.: Carbonaceous and ionic compositional patterns of fine particles over an urban Mediterranean area, Sci. Total Environ., 424, 251–263, 2012.

Theodosi, C., Grivas, G., Zarmpas, P., Chaloulakou, A., and Mihalopoulos, N.: Mass and chemical composition of size-segregated aerosols (PM1, PM2.5, PM10) over Athens, Greece: local versus regional sources, Atmos. Chem. Phys., 11, 11895-11911, htt

Please also note the supplement to this comment:
https://www.atmos-chem-phys-discuss.net/acp-2018-356/acp-2018-356-AC1-supplement.pdf
* * *

---

## Author Comment (AC2) · 23 Aug 2018

Response to Anonymous Referee #2 comments

This manuscript presents a long-term dataset of near real time chemical composition of submicron aerosols in Athens, Greece. It is completed by two intensive campaigns

during winter time. Statistical analysis was performed in order to apportion the sources of organic matter. The subject of this paper is of interest and falls within the scope of ACP, although in its current form neither the methodology (PMF) nor the results bring strikingly new outputs in this region. I am still favorable to publication after major revisions.

Response: We thank the anonymous referee for the review and have incorporated his/her suggestions and comments in the revised version of the manuscript.

**Major Comments** 1) Overall, the result section is too descriptive. Describing the angles between profiles impinges upon the actual results. The authors should re-focus the discussions on how this study slots into previous knowledge in Greece (Athens and their cities, such as Patras) and, why not, in the Eastern Mediterranean.

Response: We would like to thank the reviewer for this comment. Given the fact that matters of clarity and focus have been also pointed out by some of the other reviewers, the results section and especially §3.3, has now been revised in the new version of the manuscript. Similarity with literature spectra is now presented in a more systematic and clear way, while the discussion incorporates a more in depth analysis of the novel findings that complement the knowledge obtained regarding submicron aerosols and more specifically the organic fraction, for the region of the Eastern Mediterranean.

Moreover, strong assessments are made regarding SOA, simply from diurnal variations. The authors should either tone down these statements or add much more discussion and figures.

Response: We do not feel that our assessment of the nature/origin of SOA is simply based on diurnal variability. SV-OOA which is believed to originate from the fast oxidation of primary combustion sources is correlated during winter-time both with external tracers such as BCwb and fine-mode nss-K+, as well as with mass spectra from oxidized biomass burning. Additionally, the contribution of m/z=60 and m/z=73, well known fragments of levoglucosan, is significant to this secondary factor. Additional in-

formation for compounds such as VOCs would be useful, but this is outside the purpose of this manuscript. Nevertheless, the fast oxidation of fresh biomass burning in plumes within just a few hours after emission has been documented in literature (Lathem et al., 2013; Cubison et al., 2011), supporting our present assessment. Furthermore, the LV-OOA spectrum exhibits a highly oxidized nature, correlating very well with literature spectra of OOA attributed to regional processes from earlier studies in the region (Bougiatioti et al., 2014). In any case, we have tried in the revised version to tone down the statements, where not fully supported by experimental results.

Then, organonitrates have been found to have significant contributions in Greece (Florou et al., 2017). Can the authors add some more knowledge about that? (especially regarding the role of primary combustion sources in their formation?)

Response: We would like to thank the reviewer for pointing out the issue of organic nitrate contribution to nitrate measured by the ACSM, a matter of rising concern among the scientific community. For pure ammonium nitrate, Fry et al. (2009) report a NO+:NO2+ ratio of 2.7, while Farmer et al. (2010) report 1.5. Older studies using AMS, report ratios ranging from 1.18 (Cottrell et al., 2008) to 2 or 3 (Alfarra et al, 2006), while more recently Kindler-Scharr et al. (2016) report values ranging from 2.04 to 3.45. Using data from the ammonium nitrate calibration of our instrument we calculate a NO+:NO2+ ratio of 3.84, slightly higher than those presented in the literature. Given that the NO+:NO2+ ratio highly depends on each instrument and it's tuning, and also due to the fact that the instrument's response - in terms of this ratio - when sampling specific organic nitrate standards was not explored during this study, some uncertainty is expected when calculating the fraction of nitrate signal attributed to organonitrates (RON). These uncertainties are expected to be even larger since the limited sensitivity of the instrument does not allow to distinguish the interference of the CH2O+ ion at m/z=30, thus potentially leading to an overestimation of the measured NO+:NO2+ ratio. In our study, for the 2016-2017 period the NO+:NO2+ ratio ranged roughly from 3.20 (10th percentile) up to 9.83 (90th percentile), thus yielding several negative values for RON and a few values larger than 1, when calculated according to Farmer et al. (2009). Eventually the RON data acquired had to be treated so that negative values correspond to a fractional contribution of organonitrates to NO3 signal of 0, meaning that we assume that those negative values correspond to pure ammonium nitrate particles. Furthermore, we assume that larger values of NO+:NO2+responsible for elevated RON values are not due to other inorganic salts such as NaNO3 and Ca(NO3)2. In the above context we find that during the cold season RON is pretty low with an average value of 0.19. Diurnal variability during the cold season shows minima during night time while a maximum is evident in the afternoon. The opposite trend is observed in the warm season were RON averages at 0.62, exhibiting maxima during night time. Given all the points presented above we believe that although organonitrates is a very important subject, we feel that first it is beyond the scope of this manuscript and second that the overall uncertainties of the ACSM do not allow an accurate and qualitative determination of their levels in the present work.

2) The method used by the authors to select the appropriate solution is not clearly stated, although it was inspired by Crippa et al. (2014). From the first ACSM intercomparison, Frohlich et al. (2015) proposed a methodology of find optimal solution, and stated some recommendations. Why did the authors prefer Crippa et al.?

Response: We agree with the anonymous referee that the presentation of the PMF strategy used, which is indeed the one proposed by Crippa et al. (2014), has to be described in a more clear and systematic way. In the original manuscript the results of each step of the process were not presented, in an effort to keep the manuscript short and easy to follow. Both those issues are addressed in the revised version. . The steps of the strategy are clearly presented and PMF solutions of a) the unconstrained runs, b) runs with a constrained HOA factor and c) runs with both HOA and BBOA factor constrained for the cold season dataset are now presented in the revised supplement. Regarding the methodology selected, we believe that Frohlich et al. (2015) performed such a detailed optimization of the alpha values used to constrain each factor, due

to the scope of their work, which was specifically to explore the performance of the ME-2 method itself across different instruments, thus trying to assure a high level of compatibility amongst 15 different ACSM datasets. However, in order to fully address the reviewer's comment, the revised version of the manuscript and supplement now incorporates results from a sensitivity analysis performed on the alpha value approach.

3) The local vs regional vs advected features are not well characterized, and I would strongly recommend the authors to perform a wind analysis, especially for the "local" sources (eg traffic & biomass burning).

Response: According to the reviewer's suggestion, wind effects have been examined in the revised manuscript, focusing on the sources of local character, namely traffic and biomass burning, as expressed by the HOA and BBOA time-series. The concentrations of these parameters, when regressed against wind speed, decline exponentially for winds exceeding 2 m sec-1, indicating the locality of the respective sources (Fourtziou et al., 2017). The examination of bivariate wind direction and speed plots (Grivas et al., 2018) further corroborate that organic aerosols characterized as HOA and BBOA are mainly produced in the vicinity of the sampling location, rather than advected from the extended area of Athens. This information, is included in the revised manuscript, accompanied by relevant supplementary figures.

**Minor comments** P3 l78: the introduction mainly focuses on wintertime biomass burning, so why would you need long-term datasets?

Response: The introduction has now been revised so as to be much more balanced and provide discussion also related to the warm periods of the year. Furthermore, the importance of long-term chemical composition datasets with high temporal resolution, in the urban setting, is highlighted.

P4 l120: please indicate the calibration values

Response: Calibration values have been added to the supplement.

P4 l103: the ACSM does not measure "aerosol mass" but only the chemical composition; it is not equivalent to a TEOM-FDMS.

Response: This phrasing has been corrected in the revised manuscript.

P5 l123: it is not clear why a chemical-dependent CE has not been applied. Although it is discussed later on, it could be quickly stated here.

Response: We would like to thank the reviewer for pointing this out. A chemical dependent CE according to Middlebrook et al., 2011, has been now applied on the dataset and will be incorporated in the revised version. Concentrations have been updated accordingly.

P5 l125: did the authors use denuders ahead of the PILS in order to prevent nitric acid, sulfuric acid and ammonia to be respectively confused with particulate nitrate, sulfate and ammonium?

Response: Two denuders are placed in front of the PILS inlet, which is now added as information in the revised version of the manuscript. Note also that PILS was operating in cation mode and mainly nss-K were used in this manuscript.

P5: filter samplings are not presented.

Response: Details on filter sampling and analysis has been added.

P10 l292: See major comment. I don't think that the only fact that nitrate has a morning peak similar to BC is enough to link it with morning traffic. More discussion would be needed.

Response: We agree that a direct comparison between nitrate and BC is not enough to link with morning traffic. Nitrate is an oxidation product, probably what we see is the combined result of several factors like nighttime oxidation, pH, BL. and therefore we have decided to remove this sentence from the revised text.

P11 l318-319: Datasets have been separated into cold and warm months prior to PMF.

Why not seasonally? Not just cold and warm months influence the characteristics of secondary organic aerosols, it could also be related to air masses. So the approach chosen here is not well justified.

Response: According to studies on the climatology of Southern Greece, the transient period (spring and fall seasons) in Athens doesn't exceed 60 days on average (Argyriou et al., 2004), covering mainly the months of April and October - which were excluded from the seasonal analysis. As a result, the vast majority of air pollution studies in the area have examined the seasonal variability on a cold-warm season basis (Grivas et al., 2008 and reference therein). This classification reflects the contrast between dry/sunny summer and humid/cloudy winter conditions which are typical in the Mediterranean climate setting. Moreover, this split is representative of the relative intensity of local sources and processes, since the cold period effectively includes the time-span when residential heating is active as a source. On the other hand, the warm period essentially coincides with the photochemical season, also being characterized by an absence of precipitation and therefore a larger impact from regional sources.

P11 l329: why the HOA from Ng et al. has been used? Why not any other profiles, especially gotten from previous studies in Greece?

Response: The HOA profile from Ng et al. (2011) is an average of profiles derived from different environments in Europe, Asia and North America, covering different seasons of the year and countries with different types of vehicular fleet. Consequently, we consider that the HOA factor of Ng et al. (2011) can be suitable as an anchor profile for constrained runs in the present study. The results obtained using this factor as an anchor are proven to be representative. Correlation with HOA factors from other studies in Greece is very good for all seasons. Correlation with the wintertime HOA obtained by Florou et al. (2017) at the same measuring site yields an R2=0.92 while correlation of this study's deconvolved factors for HOA during summertime of 2016 and 2017, with the HOA factor of Kostenidou et al. (2015) measured at a suburban site in Athens during the summer of 2012, was also excellent (R2=0.92 and R2=0.94

respectively). All of this information has now been added to the revised manuscript.

P11 l333: Do the authors have any hint of how representative the BBOA of Ng et al. is in Greece?

Response: We agree with the reviewer that the nature of the biomass burning fuel may vary considerably depending on the location. Especially in Athens, Fourtziou et al. (2017) have reported that a wide variety of hardwood and softwood types are commonly used for heating purposes. This is why an average BBOA spectrum has been selected, and a large alpha value imposed. The BBOA spectrum from Ng et al. (2011) exhibits very good correlation with BBOA factors identified in previous studies for two major Greek cities. Comparison with the Florou et al. (2017) BBOA factor for Athens yields an R2=0.91 while comparison with the factor for Patras yields an R2=0.90. BBOA from this study for the 2016-2017 winter, also correlates very well with R2=0.94. Finally BBOA from Ng et al. (2011) correlates well with BBOA obtained at Finokalia, a regional background site (Bougiatioti et al., 2014) yielding R2=0.78. The results obtained during this study justifies our decision, as the derived BBOA is in excellent correlation with the respective ones found in the literature. one obtained during wintertime by Florou et al. (2017) in Athens (R2=0.89), and in Patras (R2=0.89), even with the BBOA from Bologna during winter (R2=0.85; Gilardoni et al., 2016).

P12 l340-341: it could be appreciable if the authors provide a bit more details on the metrics used through the correlation of PMF timeseries with external tracers.

Response: Even though it is stated at the supplementary material, we agree with the reviewer that this piece of information was obscured by the way the correlation tables were presented. This matter has been considered in the revised version of the supplement.

P12 l351: a correlation coefficient of 0.86 corresponds to a r2 of 0.63, which is still a good statistical correlation. Ranges of r2 are by the way not consistent throughout the manuscript.

Response: The inconsistency in terms of reported correlations was also pointed out by some of the other referees. In the revised text all correlations are given in terms of the square of the Pearson correlation coefficients (R2).

P13 l397 and p14l400, r2 of 0.32, 0.36 and 0.39 are considered as "moderate", which should rather be a poor correlation. Later, p16 l477, a r2 of 0.53 is considered as "very well". I strongly suggests the authors to use a consistent description of correlation coefficients.

Response: Both of the abovementioned comments have been addressed in the revised manuscript. Characterization of the derived correlations is now uniform in the revised version. Correlations are presented in a more systematic way. Only one metric is used when investigating both affinity with other mass spectra and correlation with external time series. Quantitative criteria regarding the description of correlations are also set.

P13 l388-390: See major comment. Linking SVOOA with primary sources only from mean diurnal variations is not convincing. Please add more discussion.

Response: The phrasing of this sentence has been changed in the revised version of the manuscript, even though the assumption made, does not refer to the diurnal variability of the factor, but rather to its comparison with reference mass spectra obtained in other studies. In this context, it is now made clear, that since the warm season SV-OOA factor spectrum, obtained during this study, correlates better to factors related with the oxidation of biogenic VOCs together with SOA factors related to primary sources such as cooking and traffic an assumption linking the factor to primary sources can be made.

P14 l400: See major comment. Same comment, the statement "SVOOA may, to some extent, partially originate from a combustion source" seems random and hardly quantifiable.

Response: We agree with the reviewer that the fact that SV-OOA may originate from a combustion source is not quantifiable, but, nevertheless, it must originate from the

oxidation process of some kind of primary organic aerosol. Its good correlation with external spectra of aged combustion emissions (aged diesel exhaust, as well as aged BBOA), supports the validity of the statement.

P14 l407: "HOA emissions are very low", compared to what?

Response: This sentence should rather read "HOA concentrations are very low compared to the cold season" since HOA concentration during the warm months is on average more than five times lower than during the cold season. This information has been added to the revised manuscript.

P14 l422: the authors would need to prove the link between SOA and regional biomass burning.

Response: The link between SOA from the oxidation of primary biomass burning emissions is not new in the literature. It has been demonstrated that OOA of a mixed nature, but which is also made up of aged BBOA, has been observed in a regional background site in Spain, 100 km away from a wildfire (Minguillón et al. 2015) and also in a regional background site in Greece, affected by plumes from wildfires over even longer distances (Bougiatioti et al., 2014). With air masses during the warm period originating mostly from the north, northeastern sector, it is clear that part of the transported OOA is bound to include to some extent (nonetheless non-quantifiable) processed, regional biomass burning from the Balkans and the area of the Black Sea, where hotspots of fires are peaking during July-September (Sciare et al., 2008).

P15 l444-448: how does BBOA compare with BBOA profiles from other studies in Greece? Or in other Mediterranean sites?

Response: BBOA correlates very well with the spectra reported by Florou et al.,2017 both for the BBOA factor obtained at the same site (R2=0.88) and BBOA in Patras (R2=0.89). It also correlates well with the BBOA factor obtained at Finokalia (Bougiatioti et al., 2014) exhibiting an R2 of 0.81 as well as with the BBOA obtained in Bologna

by Gilardoni et al., 2016 with an R2 of 0.85.

P16 l477-479: HOA correlates moderately with CO, BC and NO. So is HOA representative of traffic?

Response: HOA is representative for fossil fuel combustion processes, and it may originate from both traffic and domestic heating oil emissions. Performing a linear regression exercise for the cold period data set, excluding values obtained during night-time (19:00 – 05:00) the HOA factor exhibits stronger correlation with tracers that are related to traffic emissions, than when using the entire dataset. Indicatively, during the day-time period including the traffic rush hour peak, HOA correlates well with BCff (R2=0.71).

**Technical corrections and suggestions** P1 l22: replace "fine" by "submicron" Response: Amended. P1 l24: rephrase to "with concentrations during wintertime sporadically reaching up to 200 $\mu$g/m3". Please also indicate the time resolution for this (daily/hourly concentrations?) Response: Amended. P2 l50: replace "namely" by "such as" Response: Amended. P4 l92: "105 m above sea level" Response: Amended. P4 l105-109: these information are redundant and/or well known. It could be removed.

Response: This information has been now removed.

P6 l165-172: I don't think a thorough description of PMF and ME-2 is necessary here. Please shorten or remove this section. Response: Amended P8 l237: rephrase to "The other striking feature is that" Response: Amended. P8 l241: rephrase to "average 8 of such" Response: Amended. P8 l242: please add "to our knowledge" Response: Amended. P8 l243: rephrase to "highlight the strong impact" Response: Amended. P10 l286: rephrase to "to the regional character" Response: Amended.

P11 l231: one could cite here Canonaco et al.(2015): Canonaco, F., Slowik, J. G., Baltensperger, U., and Prévôt, A. S. H.: Seasonal differences in oxygenated organic aerosol composition: implications for emissions sources and factor analysis, Atmos.

[Figure]

Chem. Phys., 15, 6993-7002, https://doi.org/10.5194/acp-15-6993- 2015, 2015. -

Response: The proposed references are now added in the revised text.

P12 l343-350: I think this has already been presented elsewhere, so I don't think it is necessary here.

Response: Given the fact that a new approach in describing correlations, has been adopted for the revised manuscript, omitting the use of the theta angle, this description has now been removed.

References

Alfarra, M. R., Paulsen, D., Gysel, M., Garforth, A. A., Dommen, J., Pr'evËE̜ot, A. S. H., Worsnop, D. R., Baltensperger, U., and Coe, H.: A mass spectrometric study of secondary organic aerosols formed from the photooxidation of anthropogenic and biogenic precursors in a reaction chamber, Atmos. Chem. Phys., 6, 5279– 5293, 2006.

Argyriou A, Kassomenos P, Lykoudis S. On the methods for the delimitation of seasons. Water Air Soil Pollut Focus, 2004;4:65–74.

Bougiatioti, A., Stavroulas, I., Kostenidou, E., Zarmpas, P., Theodosi, C., Kouvarakis, G., Canonaco, F., Prévôt, A. S. H., Nenes, A., Pandis, S. N., and Mihalopoulos, N.: Processing of biomass-burning aerosol in the eastern Mediterranean during summertime, Atmos. Chem. Phys., 14, 4793-4807, https://doi.org/10.5194/acp-14-4793-2014, 2014.

Cottrell, L. D., Griffin, R. J., Jimenez, J. L., Zhang, Q., Ulbrich, I., Ziemba, L. D., Beckman, P. J., Sive, B. C., and Talbot, R. W.: Submicron particles at Thompson Farm during ICARTT measured using aerosol mass spectrometry, J. Geophys. Res.-Atmos., 113, D08212, doi:10.1029/2007JD009192, 2008.

Crippa, M., Canonaco, F., Lanz, V. A., Äijälä, M., Allan, J. D., Carbone, S., Capes, G., Ceburnis, D., Dall'Osto, M., Day, D. A., DeCarlo, P. F., Ehn, M., Eriksson, A., Freney,

[Figure]

E., Hildebrandt Ruiz, L., Hillamo, R., Jimenez, J. L., Junninen, H., Kiendler-Scharr, A., Kortelainen, A.-M., Kulmala, M., Laaksonen, A., Mensah, A. A., Mohr, C., Nemitz, E., O'Dowd, C., Ovadnevaite, J., Pandis, S. N., Petäjä, T., Poulain, L., Saarikoski, S., Sellegri, K., Swietlicki, E., Tiitta, P., Worsnop, D. R., Baltensperger, U., and Prévôt, A. S. H.: Organic aerosol components derived from 25 AMS data sets across Europe using a consistent ME-2 based source apportionment approach, Atmos. Chem. Phys., 14, 6159-6176, https://doi.org/10.5194/acp-14-6159-2014, 2014.

Cubison, M. J., Ortega, A. M., Hayes, P. L., Farmer, D. K., Day, D., Lechner, M. J., Brune, W. H., Apel, E., Diskin, G. S., Fisher, J. A., Fuelberg, H. E., Hecobian, A., Knapp, D. J., Mikoviny, T., Riemer, D., Sachse, G. W., Sessions, W., Weber, R. J., Weinheimer, A. J., Wisthaler, A., and Jimenez, J. L.: Effects of aging on organic aerosol from open biomass burning smoke in aircraft and laboratory studies, Atmos. Chem. Phys., 11, 12049–12064, doi:10.5194/acp-11-12049-2011, 2011.

Farmer, D.K., Matsunaga, A., Docherty, K.S., Surratt, J.D., Seinfeld, J.H., Ziemann, P.J. and Jimenez, J.L., 2010. Response of an aerosol mass spectrometer to organonitrates and organosulfates and implications for atmospheric chemistry. Proceedings of the National Academy of Sciences, 107(15), pp.6670-6675.

Florou, K., Papanastasiou, D. K., Pikridas, M., Kaltsonoudis, C., Louvaris, E., Gkatzelis, G. I., Patoulias, D., Mihalopoulos, N., and Pandis, S. N.: The contribution of wood burning and other pollution sources to wintertime organic aerosol levels in two Greek cities, Atmos. Chem. Phys., 17, 3145-3163, https://doi.org/10.5194/acp-17-3145-2017, 2017.

Fourtziou, L., Liakakou, E., Stavroulas, I., Theodosi, C., Zarmpas, P., Psiloglou, B., Sciare, J., Maggos, T., Bairachtari, K., Bougiatioti, A. and Gerasopoulos, E., 2017. Multi-tracer approach to characterize domestic wood burning in Athens (Greece) during wintertime. Atmospheric Environment, 148, pp.89-101.

Fröhlich, R., Crenn, V., Setyan, A., Belis, C. A., Canonaco, F., Favez, O., Riffault, V.,

Slowik, J. G., Aas, W., Aijälä, M., Alastuey, A., Artiñano, B., Bonnaire, N., Bozzetti, C., Bressi, M., Carbone, C., Coz, E., Croteau, P. L., Cubison, M. J., Esser-Gietl, J. K., Green, D. C., Gros, V., Heikkinen, L., Herrmann, H., Jayne, J. T., Lunder, C. R., Minguillón, M. C., Močnik, G., O'Dowd, C. D., Ovadnevaite, J., Petralia, E., Poulain, L., Priestman, M., Ripoll, A., Sarda-Estève, R., Wiedensohler, A., Baltensperger, U., Sciare, J., and Prévôt, A. S. H.: ACTRIS ACSM intercomparison – Part 2: Intercomparison of ME-2 organic source apportionment results from 15 individual, co-located aerosol mass spectrometers, Atmos. Meas. Tech., 8, 2555-2576, https://doi.org/10.5194/amt-8-2555-2015, 2015.

Fry, J. L., Kiendler-Scharr, A., Rollins, A. W., Wooldridge, P. J., Brown, S. S., Fuchs, H., Dubé, W., Mensah, A., dal Maso, M., Tillmann, R., Dorn, H.-P., Brauers, T., and Cohen, R. C.: Organic nitrate and secondary organic aerosol yield from NO3 oxidation of $\beta$-pinene evaluated using a gas-phase kinetics/aerosol partitioning model, Atmos. Chem. Phys., 9, 1431-1449, https://doi.org/10.5194/acp-9-1431-2009, 2009.

Gilardoni, S., Massoli, P., Paglione, M., Giulianelli, L., Carbone, C., Rinaldi, M., Decesari, S., Sandrini, S., Costabile, F., Gobbi, G.P. and Pietrogrande, M.C., 2016. Direct observation of aqueous secondary organic aerosol from biomass-burning emissions. Proceedings of the National Academy of Sciences, 113(36), pp.10013-10018.

Grivas, G., Cheristanidis, S., Chaloulakou, A., Koutrakis, P., and Mihalopoulos, N.: Elemental composition and source apportionment of fine and coarse particles at traffic and urban background locations in Athens, Greece, Aerosol Air Qual. Res., 18, 1642-1659, 2018

Grivas, G., Chaloulakou, A. and Kassomenos, P., 2008. An overview of the PM10 pollution problem, in the Metropolitan Area of Athens, Greece. Assessment of controlling factors and potential impact of long range transport. Science of the total environment, 389(1), pp.165-177.

Kiendler‐Scharr, A., Mensah, A.A., Friese, E., Topping, D., Nemitz, E., Prevot,

A.S.H., Äijälä, M., Allan, J., Canonaco, F., Canagaratna, M. and Carbone, S., 2016. Ubiquity of organic nitrates from nighttime chemistry in the European submicron aerosol. Geophysical Research Letters, 43(14), pp.7735-7744.

Kostenidou, E., Florou, K., Kaltsonoudis, C., Tsiflikiotou, M., Vratolis, S., Eleftheriadis, K., and Pandis, S. N.: Sources and chemical characterization of organic aerosol during the summer in the eastern Mediterranean, Atmos. Chem. Phys., 15, 11355-11371, https://doi.org/10.5194/acp-15-11355-2015, 2015.

Lathem, T. L., Beyersdorf, A. J., Thornhill, K. L., Winstead, E. L., Cubison, M. J., Hecobian, A., Jimenez, J. L., Weber, R. J., Anderson, B. E., and Nenes, A.: Analysis of CCN activity of Arctic aerosol and Canadian biomass burning during summer 2008, Atmos. Chem. Phys., 13, 2735–2756, doi:10.5194/acp-13-2735-2013, 2013.

Middlebrook, A.M., Bahreini, R., Jimenez, J.L. and Canagaratna, M.R., 2012. Evaluation of composition-dependent collection efficiencies for the aerodyne aerosol mass spectrometer using field data. Aerosol Science and Technology, 46(3), pp.258-271.

Ng, N.L., Canagaratna, M.R., Jimenez, J.L., Zhang, Q., Ulbrich, I.M. and Worsnop, D.R., 2010. Real-time methods for estimating organic component mass concentrations from aerosol mass spectrometer data. Environmental science & technology, 45(3), pp.910-916.

Sciare, J., Oikonomou, K., Favez, O., Liakakou, E., Markaki, Z., Cachier, H., and Mihalopoulos, N.: Long-term measurements of carbonaceous aerosols in the Eastern Mediterranean: evidence of long-range transport of biomass burning, Atmos. Chem. Phys., 8, 5551-5563, https://doi.org/10.5194/acp-8-5551-2008, 2008.

Please also note the supplement to this comment:
https://www.atmos-chem-phys-discuss.net/acp-2018-356/acp-2018-356-AC2-supplement.pdf

**[ACPD](ACPD)**

Interactive
comment

---

## Author Comment (AC3) · 23 Aug 2018

Atmos. Chem. Phys. Discuss., https://doi.org/10.5194/acp-2018-356-RC1, 2018 © Author(s) 2018. This work is distributed under the Creative Commons Attribution 4.0 License.

Response to Anonymous Referee #3 comments

The manuscript "Sources and processes that control the submicron organic aerosol in an urban Mediterranean environment (Athens) using high temporal resolution chemical composition measurements" presents the submicron aerosol chemical composition in Athens, Greece. In addition to study the seasonal variation of the main chemical species, organics, sulfate, nitrate, ammonium, chloride and black carbon, the sources of organics were assessed by statistical methods using Positive Matrix Factorization (PMF). The results of PMF showed that in winter there were five factors for organic aerosol (OA); fossil fuel combustion (HOA), biomass burning (BBOA), cooking (COA) and two different oxidized organic aerosols (SV-OOA and LV-OOA), of which primary sources were pronounced. In summertime, most of the OA was associated with oxidized factors representing secondary organic aerosol. This paper exploits an extensive data set (more than a year of data) and the instruments used are present-day. However, the results of this study follow very closely to those presented previously for urban areas in winter and summer not revealing any novel sources of aerosols or phenomena in urban area.

My main concern is though the PMF/ME2 analysis. Authors found biomass burning and cooking factor by constraining them with reference mass spectra. My feeling is that any factor can be constrained and a mass fraction of _10% is obtained for that factor even though there is no clear evidence of the existence of that factor. A standardized methodology to perform source apportionment on AMS data using the ME2 is given in Crippa at al. (2014) but since the authors do not show the results (residues) without constraining factors, or constraining only HOA, I can't be sure that the given methodology has been followed. My fear is that authors discovered factors that do not exist (especially COA). As it is discussed Mohr et al. (2012) the actual differentiation between AMS aerosol spectra from cooking and traffic (or BBOA) is difficult for unit mass resolution spectra (ACSM data), and it is mostly based on the relative abundances of signals at m/z 55 and 57. Authors need to provide the evidence of COA more carefully. According to Crippa et al. (2014) the presence of the meal hour peaks is necessary to support COA at least in urban areas. In the paper of Stavroulas et al. it is stated that COA exhibits a slight hump during lunchtime but this hump is very difficult to see from the figures. COA as well as all the other PMF factors, except LV-OOA, had

largest concentrations in nighttime. If meteorology (boundary layer height) affects that much on concentrations, PMF analysis can be very tricky and it may not be possible to distinguish all the sources, and that needs to be acknowledged in the paper. I think that the data presented in this paper in worth publishing. However, major changes need to be done before this paper merits publication in ACP. I recommend that authors redo PMF analysis according to Crippa et al. (2014) and consider the validity of BBOA and COA in every step (and show results from every step in supplement). Additionally, I suggest authors to concentrate on novel results that interest the whole scientific community not just Athens area, and state it clearly what are the new findings presented in this paper.

Response: We thank the anonymous referee for the review and we try to incorporate his/her suggestions and comments in the revised version of the manuscript. Authors have taken into consideration the referee's concerns, especially concerning the PMF analysis and the COA factor and have addressed the raised issues respectively. The revised manuscript includes a clearer approach in presenting the different PMF runs, reporting all the steps of the strategy and the evaluation of the results in a systematic way. A sensitivity analysis of the alpha values used is also presented, depicting the validity of the derived factors (e.g. replicating the methodology of Mohr et al. (2012)).

Major comments 1. Page 2-3, Introduction; Introduction section concentrates too much on Athens area and do not give general introduction to the research questions and issues related. I suggest taking more global point of view to the topic in introduction.

Response: The revised version the Introduction is more focused on the novel results that interest the wider scientific community and not only on wintertime emissions and biomass burning.

2. Page 11; "3.3. Source apportionment of organic aerosol" section is too long. Because the methods (PMF/ME2) are quite commonly used nowadays, and described in the literature, this section needs to be shorten or moved to experimental or supplement

leaving only clear results to "Results and Discussion" section. Authors used ME2 traditional way so there are no scientifically new results in this section regarding the use of ME2.

Response: The approach on presenting the PMF analysis has been updated in the revised version, since most reviewers suggested clarifications in the apportionment strategy followed and the presentation of the results. However, following the reviewer's request, it was moved in the supplementary material. The section features now less discussion on the PMF method and focuses in a more efficient way to the results.

3. Page 11, line 323; unconstrained runs, the results from unconstrained runs need to be presented in supplement. It is very difficult for the reader to trust the results (especially BBOA and COA factors) if unconstrained results are not shown. The technical guidelines for constraining are given in Crippa et al. (2014) and the results for each step needs should be presented.

Response: The results from the unconstrained runs, together with runs with only one factor constrained with a reference HOA spectra, and additional runs with two factors constrained with HOA and BBOA reference spectra have been added to the supplement. It is, in our opinion, evident that in the unconstrained runs the HOA-like factor is present. When constraining the HOA factor and observing the spectra in the 4 and 5 solution runs, factors clearly resembling to COA and BBOA emerge.

4. Page 12; affinity between spectra by the theta angle approach, why did you use this approach here and Pearson correlation (with R2 earlier)? It is very confusing for readers that are not familiar with this angle approach. I suggest to use Pearson correlations (R2) throughout the manuscript.

Response: We used the theta angle approach for further justification of the selected solutions. As stated in the manuscript (L348) spectra with angles larger than 30 degrees correspond to correlation coefficients <0.86, which can still be considered as a strong correlation, even though as a theta angle it is considered that the spectra

exhibit significant differences. As the confusion for readers that are not familiar with this approach has also been pointed out by some other reviewers, squared Pearson correlations (R2) are now used in the revised version of the manuscript.

5. Meteorological parameters; meteorological parameters are not given in the paper. Please provide at least temperature, radiation and boundary layer height that are important regarding the concentrations and the sources of aerosol.

Response: Meteorological parameters of ambient temperature, relative humidity, solar radiation and wind speed are now incorporated in the revised supplementary material. Unfortunately, measurements of PBL height were not available during the reported periods. The discussion regarding PBL height has been updated though and is going in to more detail, given the fact that long term observations of PBL height in Athens have been recently published (Alexiou et al., 2018).

Minor comments 6. Page 1-2, Abstract; line 30-31; "These results highlight the rising importance of biomass burning in urban environments during wintertime." The contribution of biomass burning to organics was 10% in wintertime. It's quite a small contribution. This sentence needs an evidence or to be modified.

Response: It is true that the contribution of the primary biomass burning OA factor is around 10% of organic aerosols. Nevertheless, the SV-OOA factor identified for the winter period contributes another 31% and is strongly linked to biomass burning as indicated by its correlation with tracers such as BCwb (R2=0.85) and nss-K + (R2=0.61) elevating the biomass burning related factors contribution to 41% of OA.

7. Page 3, line 82; "non-refractory part"; you also measured BC, why it is not included in main objectives (BC is refractory component)?

Response: The term non-refractory part is now omitted.

8. Page 4, line 101-102; "s/n 140-139" not needed here

Response: The instrument's serial number has been removed from the revised

manuscript

9. Page 4, line 102; Aerodyne Research Inc.

Response: Amended

10. Page 4, lines 112-120; "The instrument has participated in an intercomparison study:" This information is not relevant. Please remove this intercomparison section or move it to supplement.

Response: Information related to the instruments participation to ACMCC's intercomparison study has now been moved to the supplement.

11. Page 4, line 118-120; give RIE values

Response: IE for NO3 and RIEs for NH4 and SO4 are now stated in the revised supplement.

12. Page 5, line 122-123; default collection efficiency of 0.5, please use equation of Middlebrook et al., (2012) to calculate composition dependent collection efficiency.

Response: The issue of using a constant collection efficiency of 0.5 has also been raised by Anonymous Referee #2. Chemical composition dependent CE has now been applied to the dataset according to Middlebrook et al. (2012). All concentrations have been updated accordingly.

13. Page 5, line 138-139; more information is needed on SMPS measurements; size range, how number size distribution was converted to mass concentration (density)?

Response: The details regarding the SMPS size range as well as the reference to the method used to convert Volume concentration obtained by the SMPS to mass concentration were given in the original manuscript in lines 210 – 217 in §3.1. This piece of information has now been moved to the more appropriate "Instruments and Methods" section.

14. Page 5, line 140-144; give more details of selected absorption exponents, are they default values or did you calculate them specifically from this data set/ for this location?

Response: The absorption exponents used for the BC source apportionment are the default values used by the AE-33 software, aff = 1 and abb = 2. No fine tuning of the apportionment model was conducted in this study. We feel that a sensitivity analysis of Angstrom exponents used, does not lie within the scope of this manuscript and will certainly be addressed in future work.

15. Page 5, line 144; remove "Necessary" Response: Amended.

16. Page 5, line 145; remove "historic" Response: Amended.

17. Page 6, line 160; on the organic mass spectra obtained Response: Amended.

18. Page 7, line 185; "following section"; give the number of sections Response: Amended.

19. Page 7, line 194-196; describe PM2.5 filter collection and thermal-optical method in experimental section

Response: Details on the filter sampling procedure and thermal-optical protocol used are now given in the revised version of the manuscript.

20. Page 8, line 223; add time base for averages e.g. 1-hour average Response: Amended.

21. Page 8, line 244, change "to the levels" to "on the levels" Response: Amended.

22. Page 8-9, line 244-247; "These observations are in accordance:" this sentence is unclear and needs to be modified

Response: This sentence has now been modified accordingly. It is now clear that the levels of maximum concentrations measured by Florou et al., 2017, during a campaign from 10/01/2013 until 09/02/2013 for organics, BC and nitrate are similar to the ones

measured for all three winter measurement periods reported in this study.

23. Page 9, line 261-262; "additional primary emissions from heating play a role", based on what? Explain how you see this addition in results.

Response: The stated addition refers to the largely elevated concentration levels of organics and BC which during winter are also emitted from central heating systems and fireplaces. Necessary clarifications have been made in the revised text.

24. Page 9, line 273; what are increased local sources for nitrate in winter?

Response: As seen in our study, but also in the study of Florou et al. (2017), nitrate concentrations follow a similar trend with the organic aerosol, as well as with BC. Therefore, the combination of the low temperatures during nighttime along with the increased local combustion sources which lead to reduced acidity, result at the favorable partitioning of nitrate in the aerosol phase. This is clarified in the revised text.

25. Page 11, line 309-312; "higher organics concentration during early night could possibly be due to biogenic/vegetation sources that produce volatile components that condenses on particulate phase during night." This assumption needs evidence, maybe reference or can you see this in mass spectra of organics?

Response: This assumption is backed when taking into account the PMF analysis of section §3.3. It is clear in Figure 5 that more than one third of the night time peak is attributed to the SV-OOA factor. SV-OOA exhibits good correlation with reference mass spectra obtained for SOA linked to the oxidation of biogenic precursors. An $R^2=0.90$ was found when correlating to IEPOX-OA from Budisulistiorini et al. (2013), while correlation with SOA formed by the oxidation of b-pinene (Bahreini et al., 2005) yielded an $R^2=0.89$. A reference to this discussion in section 3.3 has now been added to the sentence.

26. Page 12, line 354-356; if HOA; COA; SV-OOA and LV-OOA are mentioned here for the first time the long names should be given. Please double-check when abbreviations

are given for the first time.

Response: Indeed, the factor abbreviations are given here for the first time in the manuscript. In the revised text their whole names will be given with the abbreviations in parenthesis, to be used throughout the text.

27. Page 13, line 383-385; "OA precursors are maximum during night similar to SVOOA". Please give reference or results.

Response: Biogenic SOA precursors such as a- and b- pinene and limonene are known to exhibit maximum concentrations during nighttime (e.g Harrison et al., 2001 for measurements performed in a forest area of Greece). Measurements of biogenic SOA precursors performed in Athens during winter and summer time (Kaltsonoudis et al., 2017) also show an increase during nighttime. Furthermore, based on the recent field study of Li et al. (2018), isoprene-derived SOA tracers, such as Methyltetrahydrofuran-diols and C5-alkene triols, mainly formed by reactive uptake of IEPOX, exhibit a clear diurnal variability with maximum values during nighttime and minimum during day. Similarly, organosulfates derived from isoprene have been found to exhibit their higher concentrations during night, in biogenically influenced urban regions Hatch et al. (2011). All these references are now added in the text.

28. Page 13, line 385-387; "SV-OOA shares some similarities with SOA from diesel exhaust". This is too vague. Give correlation coefficient or remove sentence. How much diesel vehicles there are in Athens?

Response: A correlation coefficient for the comparison with the mass spectra obtained by Sage et al. (2008) has been added to the revised version of the manuscript. A concrete number for the diesel vehicles in Athens is not available. According to local authorities, the number of new diesel passenger vehicles sold in Greece since the lift of the long-standing ban in the two major cities exceeds 300,000. The larger part of these vehicles are expected to circulate in the area of Athens.

29. Page 13, line 3963-397; "COA shown moderate correlation with nitrate". Explain why.

Response: COA is not expected to be a semi-volatile component, therefore a correlation with nitrate is not to be expected. Probably this moderate correlation is due to the similar diurnal variability of the two components, characterized by pronounced nighttime peaks.

30. Page 14, line 403; Is figure number here really 8? Double-check figure numbers.

Response: Yes, the Figure number here is stated correctly. Figure 8 shows the results of the back-trajectory cluster analysis carried out.

31. Page 14, line 410; "COA exhibits a slight hump during lunch hours." I really can't see this hump in Figure 5. There is similar lump between 4 and 9 am. How do you explain this morning lump? Please add negative standard deviations to Figure 5 (and all the other figures as well) because it's confusing (and maybe misleading) when only positive deviations are shown. Add also zero-lines to Figure 5 and Figure 6.

Response: Performing a similar exercise as for the inorganic components diurnal variability, where we calculated the diurnal variability normalized to the mean value, it is evident that during early morning, namely at 05:00 and 06:00 the COA factor concentration is 47% and 49% of the mean daily value respectively. On the contrary during early afternoon at 13:00, 14:00 and 15:00 concentration rises to 64%, 62% and 63% of the mean value respectively.

32. Page 14, line 417; "moderate hump for SV-OOA during mid-day". I can't see this hump in Figure 5. If you think this "hump" is true show it with numbers e.g. how much SV-OOA increased during mid-day compared to e.g. morning.

Response: Following a constant decline of the, normalized to the daily mean, concentration of the SV-OOA factor, from roughly 137% of the mean to 80% starting from 00:00 until 10:00, a plateau is observed with almost constant normalized values around

80% from 10:00 to 14:00. Another decline follows until the minimum (60% of the mean) at 18:00 before the rise until midnight. Peak values of the normalized LV-OOA concentrations occur within this 10:00 to 14:00, namely 107% and 106% of the mean at 13:00 and 14:00 respectively.

33. Page 16, line 463; How did you calculate Nss-K?

Response: nss-K+ concentrations are derived from fine mode potassium measured by PILS (sampling through a PM1 cyclone) corrected for seawater influence, using Na+ concentrations (also measured by PILS) and the Na/K ratio in seawater as reference (Seinfeld and Pandis, 1998; Sciare et al., 2005). A clarification is now added in the revised version of the manuscript.

34. Page 16, line 467-471; "SV-OOA mass spectra includes also fingerprint fragments of biomass burning m/z 60 and 73"; what fraction of these mass fragments were associate with BBOA and SV-OOA (and other factors)?

Response: For the fragment m/z=60, 41.4% is attributed to the BBOA factor, 46.7% is attributed to the SV-OOA factor and 3.4%, 8.5% to HOA and COA respectively. For fragment m/z=73 the fractions are 26.6% for BBOA, 48.3% for SV-OOA, 7.1% for HOA, 12.9% for COA and 5.1% for LV-OOA.

35. Page 16, line 477-478; why COA correlates with potassium and chloride?

Response: As seen from the comparison with external mass spectra, COA is very similar (R2=0.93) to the meat charbroiling spectra found in the chamber study by Kaltsonoudis et al. (2017), therefore it is expected to exhibit a good correlation with potassium, a tracer for biomass burning from restaurants/rotisseries. Similarly, chloride is also emitted during biomass burning (Akagi et al., 2011). This reference is added to the revised text.

36. Page 16, line 484-490; "SV-OOA in cold period is linked to the fast oxidation of primary combustion sources (BBOA and HOA) which is also reflected on its diurnal

variability." This sentence needs explanation and proof.

Response: In Figure 7(b) the strong correlation of the SV-OOA factor with BCwb (R2=0.85) nss-K + (R2=0.61) and CO (R2=0.63) proposes a clear link of the semi volatile compound to primary combustion sources and especially wood burning. As this factor also correlates moderately with BCff (R2=0.40) and given that CO is also emitted by all combustion sources, a contribution from HOA oxidation cannot totally ruled out.

37. Page 17, line 494-495; "moderately hump for COA during lunchtime". This cannot be seen in Figure 6.

Response: Close attention is paid in the used terminology in the revised version of the manuscript.

38. Page 17, line 499-500 "A moderate peak during the morning traffic hour (partly masked by the high night values) for SV-OOA," This peak is very difficult to see in Figure 6 (concentrations) and it does not exist in contributions figure. Please, re-consider how you define peaks/humps etc. in the paper.

Response: As mentioned in the previous comment, we agree with the reviewer concerning the nomenclature used in the text and in the revised version we will pay attention to the terminology used.

39. Page 17, line 510-513, "SV-OOA comes from the rapid oxidation of freshly emitted BBOA", this needs more explanation. What is the oxidation process, what are the oxidants in wintertime? In general, it said that SV-OOA is linked to quick atmospheric processing of VOCs within few hours. This needs to be explained in more detail (with results).

Response: As also pointed out to anonymous referee #2, the fast oxidation of fresh biomass burning in plumes within just a few hours after emission is not new in the literature (Lathem et al., 2013; Cubison et al., 2011), and is correlated both with external

time series such as BCwb, nss-K+, and CO, as well as with mass spectra from oxidized biomass burning, therefore we do not feel that our assessments are unjustified. Although an exact mechanism is not yet established in the area, field observations suggest nighttime heterogeneous reactions and also involvement of nitrate radicals (Bougiatioti et al., 2014).

40. Page 18, line 533-534; "organics, BC and nitrate double their concentrations during night-time as a results of additional primary combustion for heating purposes." Do you suggest that nitrate and BC are mostly from heating? I think that the increase in winter in nighttime is mostly due to boundary layer change.

Response: Organics and BC are indeed mostly from heating. If the increase would mostly be due to the boundary layer, similar concentrations would also be seen during the warm season, as well, as the boundary layer height is not that different between warm and cold season (Alexiou et al., 2018). The significant enhancement of wintertime fine PM levels due to heating emissions has been well-documented for Athens, even prior to the advent of the economic recession (Chaloulakou et al., 2005). On the other hand, nitrate is not directly emitted, but the higher concentrations are due to the combined effect of temperature and reduced acidity to the partitioning in the aerosol phase, as it has already been mentioned. These are further clarified in the revised text.

41. Page 19, line 557-559; "HOA being affected by combustion from central heating", The impact of central heating was not discussed in Results section. If the authors think that this is the source of HOA it should be discussed and (justified) earlier.

Response: HOA is the factor that represents fossil fuel combustion, which is portrayed by both vehicular traffic, as well as heating oil combustion in central heating units However, based on emission inventories for Greece, primary non-methane hydrocarbon and aerosol emissions from central heating are much smaller compared to traffic (Fameli and Assimakopoulos, 2016). These points are clarified in the revised text.

42. Figure 1; Add "1-hour averaged" mass concentrations Response: Amended.

43. Figure 4; in upper figure you use "organic aerosol" but in lower figure "Organics". Please be consistent with the names. Response: Amended.

44. Figure 6; why did you plot COA and nss-K to the same figure? Based on the time series they correlate quite well. Do you suggest that they originate from the same source?

Response: The selected pairs in Figure 6 have been rearranged in order to be in accordance with those presented in figure 5.

45. Table 1; please give the name of the month clearer way e.g. using Jan, Feb etc. Response: Amended.

Technical comments: 46. Page 6, line 163; time series Response: Amended.

Rerefences

Akagi, S. K., Yokelson, R. J., Wiedinmyer, C., Alvarado, M. J., Reid, J. S., Karl, T., Crounse, J. D., and Wennberg, P. O.: Emission factors for open and domestic biomass burning for use in atmospheric models, Atmos. Chem. Phys., 11, 4039-4072, https://doi.org/10.5194/acp-11-4039-2011, 2011.Alexiou, D., Kokkalis, P., Papayannis, A., Rocadenbosch, F., Argyrouli, A., Tsaknakis, G. and Tzanis, C.G., 2018. Planetary boundary layer height variability over athens, greece, based on the synergy of raman lidar and radiosonde data: Application of the kalman filter and other techniques (2011-2016). In EPJ Web of Conferences (Vol. 176, p. 06007). EDP Sciences.

Chaloulakou, A., Kassomenos, P., Grivas, G., and Spyrellis, N.: Particulate Matter and Black Smoke concentration levels in Central Athens, Greece, Environ. Int., 31, 651–659, 2005.

Fameli, K. M. and Assimakopoulos, V. D.: The new open Flexible Emission Inventory for Greece and the Greater Athens Area (FEI-GREGAA): Account of pollutant sources

and their importance from 2006 to 2012, Atmos. Environ., 137, 17-37, 2016. Harrison, D.; Hunter, M. C.; Lewis, A. C.; Seakins, P. W.; Bonsang, B.; Gros, V.; Kanakidou, M.; Touaty, M.; Kavouras, I.; Mihalopoulos, N.; Stephanou, E.; Alves, C.; Nunes, T.; Pio, C.: Ambient isoprene and monoterpene concentrations in a Greek fir ( Abies Borisii-regis) forest. Reconciliation with emissions measurements and effects on measured OH concentrations, Atmospheric Environment, Volume 35, Issue 27, p. 4699-4711.

Jianjun Li, Gehui Wang, Can Wu, Cong Cao, Yanqin Ren, Jiayuan Wang, Jin Li, Junji Cao, Limin Zeng & Tong Zhu, 2018. Characterization of isoprene-derived secondary organic aerosol at a rural site in North China Plain with implications for anthropogenic pollution effects, Scientific Reports, 8, Article number 535 (2018).

Kaltsonoudis, C., Kostenidou, E., Louvaris, E., Psichoudaki, M., Tsiligiannis, E., Florou, K., Liangou, A., and Pandis, S. N.: Characterization of fresh and aged organic aerosol emissions from meat charbroiling, Atmos. Chem. Phys., 17, 7143-7155, doi: 10.5194/acp-17-7143-2017, 2017.

Lindsay E. Hatch, Jessie M. Creamean, Andrew P. Ault, Jason D. Surratt, Man Nin Chan, John H. Seinfeld, Eric S. Edgerton, Yongxuan Su, and Kimberly A. Prather: Measurements of Isoprene-Derived Organosulfates in Ambient Aerosols by Aerosol Time-of-Flight Mass Spectrometry—Part 2: Temporal Variability and Formation Mechanisms, Environmental Science & Technology 2011 45 (20), 8648-8655 DOI: 10.1021/es2011836. Sage, A. M., Weitkamp, E. A., Robinson, A. L., and Donahue, N. M.: Evolving mass spectra of 795 the oxidized component of organic aerosol: results from aerosol mass spectrometer analyses of aged diesel emissions, Atmos. Chem. Phys., 8, 1139-1152, doi:10.5194/acp-8-1139-2008, 2008.

Sciare, J., Oikonomou, K., Cachier, H., Mihalopoulos, N., Andreae, M. O., Maenhaut, W., and Sarda-Estève, R.: Aerosol mass closure and reconstruction of the light scattering coefficient over the Eastern Mediterranean Sea during the MINOS campaign, Atmos. Chem. Phys., 5, 2253-2265, https://doi.org/10.5194/acp-5-2253-2005, 2005.

[Figure]

Please also note the supplement to this comment:
https://www.atmos-chem-phys-discuss.net/acp-2018-356/acp-2018-356-AC3-
supplement.pdf

—————————————————

---

## Author Comment (AC4) · 23 Aug 2018

Response to Anonymous Referee #4 comments

The manuscript presents a one-year dataset (2016/2017) of near real time chemical composition of submicron aerosol particles measured in Athens and its subsequent

[Figure]

PMF analysis. This dataset is complemented by 2 intensive campaigns carried out in winter (2013/2014 and 2015/2016). While these data are of prime interest, the manuscript is very descriptive and do not bring significant new results for the scientific community. However, I support the publication of this manuscript after major modifications.

Response: We would like to thank the anonymous referee for the review and have incorporated the suggested comments in the revised version of the manuscript.

1/ The PMF analysis and the constraints applied are somewhat confusing and the methodology should be described more clearly and in a more systematical way. A lot of different alpha values are selected (arbitrarily?) for the different factors. For a given source profile authors choose different alpha values for the different dataset. This must be explained and justified. Did the authors studied the influence of the alpha values on the sources contributions in a more systematic way? An alpha value of 0.1 is, from my point of view, too low for COA. Same for HOA, an alpha value of 0.05 is, in a first approach, too low considering the variability of the vehicular fleet (diesel/gasoline share, ...).

Response: Since this has been pointed out in other reviewer reports, the revised manuscript attempts a clearer approach in presenting those results, reporting all the steps of the strategy and the evaluation of the results in a systematic way. A sensitivity analysis of the alpha values used is also presented. Affinity of the resulting factors with deconvolved factors from the literature as well as correlation of the respective time series with external tracers is the criterion of selection for the preferable PMF solution. In this manner different alpha values for the split dataset concerning the same factor shouldn't be unreasonable. Specifically, the two HOA factors obtained for the warm months of 2016 and 2017 with different a-values (0.05 and 0.1 respectively) show excellent correlation ($R2=0.99$). On the other hand, an alpha value of 0.1 for COA has been reported earlier (Cannonaco et al., 2015). Actually Crippa et al. (2014) suggest that a lower alpha value (e.g. 0.05) should be used to constrain the COA factor.

Furthermore, an alpha value in the range of 0.05 to 0.1 for HOA is suggested in the same study, and has been used in several different studies implementing the a-value approach (Frolich et al., 2015; Bressi et al., 2016).

2/ Authors should convince the reader of the validity of the COA factor extracted from their analysis. The COA factor extracted here from the PMF analysis represents a contribution as high as BBOA in winter. It seems well correlated with the BBOA factor and other combustion markers (nssK+ for instance) and do not exhibit the classical midday hump. As the COA MS profile contains a slight contribution of m/z 60, I suspect a mix of both COA and BBOA factor.

Response: It is evident, in the revised version of the supplementary material, where results from unconstrained PMF runs as well as runs only constraining an HOA factor have been added, that a factor resembling cooking-like organic aerosols, arises in both the warm and cold periods. When replicating the methodology of Mohr et al. (2012) thus calculating $f_{55,OOAsub}$ and $f_{57,OOAsub}$ and plotting them against each other, it can be observed that points lying closer to the steeper slope – obtained by fitting a line through zero and $f_{55,OOAsub}$ vs $f_{57,OOAsub}$ of deconvolved COA spectra from the literature – correspond to measurements during the afternoon (∼16:00) coinciding with this hump observed in the mean diurnal variation, and to night time measurements coinciding with the maximum observed in the diurnal variability of the factor (Figure attached).

Figure 1. f55, OOAsub plotted against f57, OOAsub. Data points are colored according to time of day. Lines correspond to linear fit results conducted using COA and HOA results both from PMF and laboratory standards studies (Mohr et al., 2012).

Lunch hours in Greece are known to be stretched towards late afternoon (15:00 – 17:00 LT). This fact has alson been demonstrated by Florou et al. (2017), where the COA factor obtained for the same site was found to be dominated by the evening peak exhibiting slightly elevated concentrations around 15:00. Note also that the same diurnal

variability and contribution of the cooking to the total mass, was reported by Florou et al., 2017 for the same site in Athens, and especially during night also in Patras, another Greek city indicating that our findings have not a local character but probably represnts all major Greek cities.

Also, the reference mass spectra chosen to constrain COA has been obtained in Paris. In Paris, the main site was located in the local Chinatown and was surrounded by well-known fast food brands. One could assume that the cooking emissions in Athens are slightly different than those of Paris for this specific study.

Response: We agree with the reviewer concerning the reference mass spectra chosen to constrain COA. The same concern was also raised by some of the other reviewers. Nevertheless, when comparing our resulting COA with the respective ones from Florou et al. (2017) (Athens, same site) and Patras (3rd largest city in Greece), the squared Pearson correlation coefficients are 0.96 and 0.93, respectively.

3/ The split of the data series between warm and cold period sounds quite arbitrary. Does it actually rely on temperature? If yes, this should be explicitly discussed in the text. While necessary for such long data series, splitting the dataset can induce a discontinuity of the sources contributions. Are such discontinuities observed here?

Response: The warm period is characterized by absence of precipitation and in-creased photochemistry which allow especially the influence from regional sources. Splitting of the data series is actually necessary when dealing with long term datasets that incorporate different sources of OA for different times of the year and splitting ap-proaches have been utilized in numerous studies that deal with long term datasets around the world (Bressi et al., 2016; Minguillon et al., 2015; Budisulistiorini et al., 2016). Given the fact that the mass spectra of the factors, present during the warm pe-riod, namely HOA, COA, SV-OOA and LV-OOA are very similar for the two parts of the warm season, e.g. July – September 2016 and May – July 2017, we can be confident that no discontinuities are present in our analysis.

4/ If the data are available, I strongly suggest that the authors carry out a local winds analysis. From my experience such high nocturnal peaks are often mostly associated to local wind changes and in this case the occurrence of nocturnal breezes. In such cases (heavily polluted urban area), a local wind analysis is, from my point of view, much more relevant than a long-range transport analysis. Also, the influence of local wind patterns can induce strong correlation within the dataset which cannot be related to sources intensities or atmospheric transformation processes.

Response: According to the suggestion of the reviewer, as well as to that of reviewer #2, a wind analysis has been performed. Although it has been verified that HOA and BBOA are related with local emissions and record high concentrations during calm and low-wind conditions, increased levels of other components, such as LV-OOA, summer-time SV-OOA and sulfate are observed during stronger winds which are linked to regional advections and long range transport. This difference, indicates that common diurnal patterns shouldn't be attributed solely to meteorological conditions. Although the reviewer is correct to indicate the importance of mesoscale circulation for receptor sites, in our case the site is located in an urban area and is additionally affected by local primary sources. Moreover, for the inner Athens basin, nocturnal land-breezes are known to be very week (Kassomenos et al., 1998) to completely determine correlation patterns due to intra-urban transport. These are now discussed in the revised version of the manuscript.

References

Bressi, M., Cavalli, F., Belis, C. A., Putaud, J.-P., Fröhlich, R., Martins dos Santos, S., Petralia, E., Prévôt, A. S. H., Berico, M., Malaguti, A., and Canonaco, F.: Variations in the chemical composition of the submicron aerosol and in the sources of the organic fraction at a regional background site of the Po Valley (Italy), Atmos. Chem. Phys., 16, 12875-12896, https://doi.org/10.5194/acp-16-12875-2016, 2016.

Budisulistiorini, S. H., Baumann, K., Edgerton, E. S., Bairai, S. T., Mueller, S., Shaw, S.

[Figure]

L., Knipping, E. M., Gold, A., and Surratt, J. D.: Seasonal characterization of submicron aerosol chemical composition and organic aerosol sources in the southeastern United States: Atlanta, Georgia,and Look Rock, Tennessee, Atmos. Chem. Phys., 16, 5171-5189, https://doi.org/10.5194/acp-16-5171-2016, 2016.

Canonaco, F., Slowik, J. G., Baltensperger, U., and Prévôt, A. S. H.: Seasonal differences in oxygenated organic aerosol composition: implications for emissions sources and factor analysis, Atmos. Chem. Phys., 15, 6993-7002, https://doi.org/10.5194/acp-15-6993-2015, 2015.

Crippa, M., Canonaco, F., Lanz, V. A., Äijälä, M., Allan, J. D., Carbone, S., Capes, G., Ceburnis, D., Dall'Osto, M., Day, D. A., DeCarlo, P. F., Ehn, M., Eriksson, A., Freney, E., Hildebrandt Ruiz, L., Hillamo, R., Jimenez, J. L., Junninen, H., Kiendler-Scharr, A., Kortelainen, A.-M., Kulmala, M., Laaksonen, A., Mensah, A. A., Mohr, C., Nemitz, E., O'Dowd, C., Ovadnevaite, J., Pandis, S. N., Petäjä, T., Poulain, L., Saarikoski, S., Sellegri, K., Swietlicki, E., Tiitta, P., Worsnop, D. R., Baltensperger, U., and Prévôt, A. S. H.: Organic aerosol components derived from 25 AMS data sets across Europe using a consistent ME-2 based source apportionment approach, Atmos. Chem. Phys., 14, 6159-6176, https://doi.org/10.5194/acp-14-6159-2014, 2014.

Florou, K., Papanastasiou, D. K., Pikridas, M., Kaltsonoudis, C., Louvaris, E., Gkatzelis, G. I., Patoulias, D., Mihalopoulos, N., and Pandis, S. N.: The contribution of wood burning and other pollution sources to wintertime organic aerosol levels in two Greek cities, Atmos. Chem. Phys., 17, 3145-3163, https://doi.org/10.5194/acp-17-3145-2017, 2017.

Fröhlich, R., Crenn, V., Setyan, A., Belis, C. A., Canonaco, F., Favez, O., Riffault, V., Slowik, J. G., Aas, W., Aijälä, M., Alastuey, A., Artiñano, B., Bonnaire, N., Bozzetti, C., Bressi, M., Carbone, C., Coz, E., Croteau, P. L., Cubison, M. J., Esser-Gietl, J. K., Green, D. C., Gros, V., Heikkinen, L., Herrmann, H., Jayne, J. T., Lunder, C. R., Minguillón, M. C., Močnik, G., O'Dowd, C. D., Ovadnevaite, J., Petralia, E., Poulain, L., Priestman, M., Ripoll, A., Sarda-Estève, R., Wiedensohler, A., Baltensperger, U., Sciare, J., and Prévôt, A. S. H.: ACTRIS ACSM intercomparison – Part 2: Intercomparison of ME-2 organic source apportionment results from 15 individual, co-located aerosol mass spectrometers, Atmos. Meas. Tech., 8, 2555-2576, https://doi.org/10.5194/amt-8-2555-2015, 2015.

Minguillón, M. C., Ripoll, A., Pérez, N., Prévôt, A. S. H., Canonaco, F., Querol, X., and Alastuey, A.: Chemical characterization of submicron regional background aerosols in the western Mediterranean using an Aerosol Chemical Speciation Monitor, Atmos. Chem. Phys., 15, 6379-6391, https://doi.org/10.5194/acp-15-6379-2015, 2015.

Mohr, C., DeCarlo, P. F., Heringa, M. F., Chirico, R., Slowik, J. G., Richter, R., Reche, C., Alastuey, A., Querol, X., Seco, R., Peñuelas, J., Jiménez, J. L., Crippa, M., Zimmermann, R., Baltensperger, U., and Prévôt, A. S. H.: Identification and quantification of organic aerosol from cooking and other sources in Barcelona using aerosol mass spectrometer data, Atmos. Chem. Phys., 12, 1649-1665, https://doi.org/10.5194/acp-12-1649-2012, 2012.

Kassomenos, P., Flocas, H.A., Lykoudis, S., and Petrakis, M.: Analysis of mesoscale patterns in relation to synoptic conditions over an urban Mediterranean basin, Theor. Appl. Climatol., 59, 215-229, 1998.

Please also note the supplement to this comment:
https://www.atmos-chem-phys-discuss.net/acp-2018-356/acp-2018-356-AC4-supplement.pdf

**Fig. 1.** f55, OOAsub plotted against f57, OOAsub. Data points are colored according to time of day. Lines correspond to linear fit results conducted using COA and HOA results both from PMF and laboratory stand

---

## Author Comment (AC5) · 23 Aug 2018

Response to Anonymous Referee #5 comments

This paper aims to identify sources of submicron organic aerosols in Athens with a major interest on quantifying the contribution of biomass burning. Results are based on

high temporal resolution chemical measurements performed by an ACSM. As stated by authors, this is the first study on submicron aerosol by using high temporal measurements during a relatively long period (1 yr plus 2 winter periods). However, it is a very descriptive work that does not provide new knowledge on atmospheric processes and sources in the eastern Mediterranean. The study is focused on the organics and mainly in the contribution of wood burning, as stated in the introduction section and as deduced from the extension of measurements during the two winter periods. Impact of wood burning in air quality is a growing concern in Athens in the last years.

Response: We would like to thank the anonymous referee for taking the time to review our manuscript. Nevertheless, we don't agree that this work does not provide new knowledge on atmospheric processes and sources in the area. For the first time such a long (20 months) high-resolution time series is presented in the literature for Athens one of the biggest cities in the eastern Mediterranean. As the reviewer claimed "Impact of wood burning in air quality is a growing concern in Athens in the last years" and indeed this was one of our aims. The final aim was to understand the sources of submicron organic aerosol in an urban location on an annual basis and not "to provide new knowledge on atmospheric processes and sources in the eastern Mediterranean". This issue which is very important necessitates measurements at multiple background sites and it was out of our scope. Note however that the Introduction was re-written to clearly address all these issues mentioned above and better highlight the importance of our work for southern European urban environments.

Thus, the authors (5 of 7) co-authored a paper currently on ACPD (https://doi.org/10.5194/acp- 2018-163) focused on the impact of residential heating on fine particulate matter by applying PMF to the chemical characterization of filters (24 and 12h resolution). This study was performed in the same place and during part of the period covered by the present study.

Response: In contrast with the study of Theodosi et al. (2018) which reports filter-based analytical data, our manuscript presents 30-min resolution on-line measurements showing in detail the diurnal variability of components, and specifically focusing on the different fractions of the organic aerosol and addresses among others the diurnal profiles of the organic fractions. Note that in the revised version of Theodosi et al., authors clearly differentiate between these two studies. Finally, less important but worth mentioning is that the two first authors are not the same in these two studies and in addition not participating in both works.

I have read the comments by the other reviewers and I strongly agree with the remarks form RC2, and also 3 and 4. I would like to add some minor comments and insist on some of the comments already mentioned by the other referees. My major concern is the use of constrains based on measurements performed in very different areas (HOA, COA and BBOA form north Europe) for the PMF of organics. Are these profiles usable in the study area? The profiles used should be more similar to the profile emissions in the area. Do the authors have some information about COA and BBOA profiles from the eastern Mediterranean area? Most statements about the origin of the SVOOA and SOA are hard to demonstrate based only in the interpretation of the diurnal variation.

Response: As also stated by the anonymous referee himself/herself most of the issues raised are common with the ones of Referees #2, #3 and #4. The revised version of the manuscript describes in detail the unconstrained and constrained runs, as well as correlations of the derived spectra of the factors with the respective ones from Greece and neighboring areas (e.g. Bologna, Italy). Furthermore, the origin of SV-OOA is clearly demonstrated for the winter period as mainly being derived from the fast oxidation of primary combustion sources, while during summer our suggestions are backed up by correlations with SOA from the oxidation of biogenic components and, to some point, regional biomass burning aerosol.

Minor comments Experimental methods; Page 5. Was the ACSM calibrated on field?

Response: The IE for NO3 and RIEs for NH4 and SO4 are now reported in the revised Supplementary material. The ACSM was calibrated just before its deployment to the

site and successfully participated in the ACSM intercomparison exercise.

No information about filters sampling and analysis is provided. Please, indicate sampling period and frequency and the methods of filters treatment and analysis.

Response: All the relevant information, sampling period, frequency and filters treatment, related to filter sampling has now been added to the revised version of the manuscript.

Please, indicate the size range of SMPS TSI3034.

Response: This piece of information was reported in section 3.1, P7, L211 of the manuscript and has been moved to Section 2.2 "Instruments and methods" of the revised version. We would like to thank the reviewer for pointing this out.

Results and discussion In the supplementary, authors show the correlations between filters and ACSM for the whole period (SL1.1) and for the winter periods (SL1.2). Is there any reason for the different slopes determined for each period? Do you expect the presence of coarse nitrate in the 1-2.5 $\mu$m fraction?

Response: We believe that the slopes for NO3-, which are 0.99 for 2016 – 2017 versus 1.02 for the winter of 2015 – 2016, the slopes for regression of NH4+ from the ACSM versus the PILS, which are 0.98 for 2016 – 2017 and 0.97 for 2013 – 2014 are not significantly different. For sulfate the small difference (0.87 for 2016 – 2017 versus 0.93 for 2015 – 2016) could be because of the different way used to calculate RIE SO4=. For the 2016 – 2017 this value is derived by calibrating the instrument with ammonium sulfate, while for 2015 – 2016 RIE was derived using the approach of Budisulistiorini et al. (2014).

Did the OA/OC ratio keep constant along the sampling period?

Response: Unfortunately, no PM2.5 filter sampling data are available for the months prior to November 2016. Furthermore, IC analysis on filters obtained through the warm period of 2017 is not yet available. In this context, even though the OM to OC ratio

varies during the cold season of 2016-17, not enough information is available in order to be able to formulate an in-depth analysis of the issue.

Did you compare EC vs BC? Is this ratio constant along the study period? Is any difference in winter with respect summer?

Response: A comparison between EC and BC hasn't been performed in the context of this study. We feel that a detailed description of the absorbing particles' behavior, falls outside the scope of this manuscript. We have to note here, that BC concentrations were used as reported by the aethalometer AE-33 deployed at the Thissio station, thus using default MAC values.

Line 256. Contribution of nitrate in summer?

Response: The contribution of nitrate during summer has been added.

Line 260. There is a BC peak in June not related to any other compound (figure 3). What is the cause of this maximum? Any information from the measurements by means of Aethalometer?

Response: When moving from May to June, an increase is also observed in organics, sulfate, and ammonium. The increase is most likely due to the relatively stagnant conditions that prevailed during June 2017 in the area (mean monthly wind speed 27% lower than the study average).

Line 265 semi-volatile inorganics; and organics?

Response: Organics have now been added to the sentence.

Lines 290 293: Is nitrate primary emitted? Do you mean that nitrate is quickly formed from primary NOx? Can be the relatively high levels of nitrate be related to the low stability of nitrate with temperature? It is risky to assign a source origin to nitrate only form the diurnal variation.

Response: This issue was also raised by referee #3. As nitrate follows the same trend

with organics and BC, it is clear that the chemical composition during nighttime is very different than the one during day. High levels during night time is certainly related to the semi volatile nature of nitrate and its partitioning between the particle and gas phase, rather than its primary emissions. The low temperatures combined with reduced aerosol acidity, favors the partitioning of nitrate in the aerosol phase.

Line 296: What do you mean with "normalizing the diurnals?

Response: The same approach is used as the one described in Lines 286 – 288. Each hourly value has been divided by the respective species mean concentration.

Line 363: Please, replace "2016 and 17" by "2016 and 2017"

Response: Amended.

Line 477. Did you check the correlation with the BC factions? Does the HOA factor correlated better with BCff than with BCwb?

Response: Yes, as can be seen in the supplementary material tables, summarizing the correlations of each selected solution with external time series, the identified HOA factors are consistently better correlated with BCff than with BCwb.

Figure 8. Why COA factor increased with wind form the eastern sector?

Response: The COA fraction for the Eastern sector in the cold period appears slightly increased (4.7% compared to the mean), due to largely decreased fractions of BBOA and SV-OOA, as compared to the Northern sectors. In fact, the mean COA concentration in the cluster, is within 0.1 $\mu$g m-3 of the mean COA concentration, a difference not statistically significant (p>0.05).

Line 480-483. During the cold period nitrate correlates with LV-OOA while in summer it correlated with SV-OOA. Could you explain the reasons of it?

Response: During the warm period nitrate correlates with SV-OOA because of the semi-volatile nature of the precursors of SV-OOA. On the other hand, during the cold

period when temperatures are lower, and because of the different chemical composition aerosol is less acidic, the low temperature and higher pH "stabilizes" more the partitioning of nitrate in the aerosol phase (Guo et al., 2017; Nah et al., 2018), therefore nitrate could correlate more with the less volatile component, which is LV-OOA.

Summary and conclusions Line 535. Sulfate and ammonium concentrations are not lower in summer

Response: This fact has now been clarified.

Line 571. What do you mean with "central heating"? Fuel-oil heating?

Response: Indeed, the fuel used in most buildings central heating installations is diesel oil. This clarification has been added in the text.

References

Budisulistiorini, S. H., Canagaratna, M. R., Croteau, P. L., Baumann, K., Edgerton, E. S., Kollman, M. S., Ng, N. L., Verma, V., Shaw, S. L., Knipping, E. M., Worsnop, D. R., Jayne, J. T., Weber, R.J., and Surratt, J. D.: Intercomparison of an Aerosol Chemical Speciation Monitor (ACSM) with ambient fine aerosol measurements in downtown Atlanta, Georgia, Atmos. Meas. Tech., 7, 1929–1941, doi:10.5194/amt-7-1929-2014, 2014.

Guo, H., Liu, J., Froyd, K. D., Roberts, J. M., Veres, P. R., Hayes, P. L., Jimenez, J. L., Nenes, A., and Weber, R. J.: Fine particle pH and gas–particle phase partitioning of inorganic species in Pasadena, California, during the 2010 CalNex campaign, Atmos. Chem. Phys., 17, 5703-5719, https://doi.org/10.5194/acp-17-5703-2017, 2017.

Nah, T., Guo, H., Sullivan, A. P., Chen, Y., Tanner, D. J., Nenes, A., Russell, A., Ng, N. L., Huey, L. G., and Weber, R. J.: Characterization of aerosol composition, aerosol acidity, and organic acid partitioning at an agriculturally intensive rural southeastern US site, Atmos. Chem. Phys., 18, 11471-11491, https://doi.org/10.5194/acp-18-11471-2018, 2018.

Theodosi, C., Tsagkaraki, M., Zarmpas, P., Liakakou, E., Grivas, G., Paraskevopoulou, D., Lianou, M., Gerasopoulos, E., and Mihalopoulos, N.: Multiyear chemical composition of the fine aerosol fraction in Athens, Greece, with emphasis on winter-time residential heating, Atmos. Chem. Phys. Discuss., https://doi.org/10.5194/acp-2018-163, in review, 2018.

Please also note the supplement to this comment:
https://www.atmos-chem-phys-discuss.net/acp-2018-356/acp-2018-356-AC5-supplement.pdf

---

## Editor Decision (ED1)

**Editor's comments on the revised version of ms. acp-2018-356 by I. Stavroulas et al., entitled "Identification of sources and processes that control submicron organic aerosol content at an urban Mediterranean environment using high temporal resolution chemical composition measurements"**

François Dulac, 16 Nov. 2018

I thank you for revision of your manuscript submitted to the ChArMEx special issue in ACP. I agree with both reviewers to acknowledge a great improvement in the manuscript. I invite you to follow their last recommendations for technical corrections and minor revisions. In the following, you fill find a list of additional technical corrections. Line numbers refer to the version of the manuscript with visible corrections included in your Authors' response. Changes are in red.

- Abstract, line 30: round up to "22%".

- Introduction, line 86: not clear to me what you mean by "precursor emissions emerge altered".

- Instruments and Methods, line 126: add a comma after "Practically".

- Instruments and Methods, line 135: Budisulistiorini et al. (2014) is absent from the reference list; do you refer to doi:10.5194/amt-7-1929-2014?

- Instruments and Methods, line 181: "were used".

- Instruments and Methods, line 188: "were also calculated".

- Instruments and Methods, line 192: "to capture".

- Source Apportionment, line 207: "of factor profiles".

- Source Apportionment, line 225: I think you should explain what is $\alpha$ here, by reintroducing the sentence that has been erased from lines 235-236 ("The $\alpha$ value ranges between 0 and 1 and is a measure of"[…]).

- Source Apportionment, line 226: "Initial" or "Initially,".

- Choosing the optimal configuration, line 274: "On the other hand, constraining two factors during the warm period, namely".

- Choosing the optimal configuration, line 276: "previous".

- Choosing the optimal configuration, line 277: "during the warm periods".

- Chemical composition and charactristics, line 352: "while maxima".

- Chemical composition and charactristics, line 354: do you mean "(up to around 100"?

- Diurnal variability, line 403: "rush hours".

- Diurnal variability, line 423: "the organics variation also follows".

- Source apportionment of organic aerosol/Warm period, l.466: "solution stems from a two factor constrained run".

- Source apportionment of organic aerosol/Warm period, l.482: "can be attributed".

- Source apportionment of organic aerosol/Warm period, l.509: remove the comma before "(2005)".

- Source apportionment of organic aerosol/Warm period, l.517: "provide some".

- Source apportionment of organic aerosol/Warm period, l.551: "when concentrations".

- Source apportionment of organic aerosol/Warm period, l.559: "night-time".

- Source apportionment of organic aerosol/Warm period, lines 575-576: "and thus exhibits".

- Source apportionment of organic aerosol/Cold period, l.578: "solution stems from a three factor constrained run".

- Source apportionment of organic aerosol/Cold period, lines 607-608: "compared to BBOA in Bologna, Athens and Patras"; provide references.

- Source apportionment of organic aerosol/Cold period, l.621: by "extended area", do you rather mean "region"?

- Source apportionment of organic aerosol/Cold period, l.622: "The identification of BBOA […] as biomass burning tracers. Indeed, BBOA exhibits excellent correlation".

- Source apportionment of organic aerosol/Cold period, l.625: "and as reported".

- Source apportionment of organic aerosol/Cold period, l.630: do you mean "and exhibits" referring to BBOA, or "which exhibits" refering to $BC_{wb}$?

- Source apportionment of organic aerosol/Cold period, lines 636-637: decapitalize "northern and eastern Europe".

- Source apportionment of organic aerosol/Cold period, l.643: relove "who " and make a new sentence after "the basin": ". They found".

- Source apportionment of organic aerosol/Cold period, l.646: change "locality"; do you mean " "?

- Source apportionment of organic aerosol/Cold period, l.650: add a comma before "respectively".

- Source apportionment of organic aerosol/Cold period, l.661: "product fraction seems" (or "products seem" but pay attention to the following "Its" and "part of it").

- Source apportionment of organic aerosol/Cold period, l.661: do you mean oxidation of "combustion emissions" rather than "combustion sources"?

- Source apportionment of organic aerosol/Cold period, l.684: "SV-OOA".

- Source apportionment of organic aerosol/Cold period, l.691: check "1-hr lag the morning traffic."

- Source apportionment of organic aerosol/Cold period, l.692: do you mean oxidation of "combustion emissions" rather than "combustion source"?

- Summary and conclusions, l.720: "in addition to routine".

- Summary and conclusions, l.743: "peak values" or "values peaking".

- Summary and conclusions, l.755: "rush hours".

- Summary and conclusions, l.773: "contribution to air quality degradation"?

- References, l.802: year at the end.

- References, lines 808-809: "Meteor. Atmos. Phys."; remove issue number "(1-3)".

- References, lines 813-814: "Environ. Sci. Technol."; remove issue number "(15)"; final dot missing.

- References, l.841: "Meas. Tech.".

- References, lines 885-886: "Atmos. Environ.".

- References, lines 893-894: "emissions, Proc. Nat. Acad. Sci., 113, 10013"; add missing doi.

- References, lines 920 and 110: remove spare lines.

- References, lines 925-926: decapitalize a number of words in the article title; "Aerosol Sci. Technol.".

- References, l.1022: "Anal. Chim. Acta"; remove issue number "(1-2)".

- References, l.1026: "data values, Environmetrics"; remove issue number "(2)".

- References, lines 1067-1069: decapitalize a number of words in the article title; "Environ. Sci. Technol., 47, 13313-13320".

- References, l.1108: "Metorol., 124, 61-79" (no italic).

- References, lines 1112-1113: "Atmos. Environ." (dots).

- References, lines 1126-1128: "Fine particulate"; "Environ. Health, 8".

- Figure 6: put the left plot above and enlarge plots as in the previous version.

- Figure 7: put the left plot above and enlarge plots.

- Figure SF.21: if possible, use a different colour or lighter green for the land background in the map, to have more contrast with the green used for SV-OOA; enlarge the text boxes associated to each pie, and the numbers within the pies, or possibly turn the figure by 90° (landscape format) and enlarge.

---

## Author Response (AR2)

**Author Response**

*We would like to thank both anonymous reviewers as well as the editor for their comments. All these comments have been taken into consideration. Below, in blue and italics, a point by point response.*

**REVIEWER #2**

The revised manuscript is of much better quality, and the authors should be acknowledged for that. They have done an accurate revision work, following the recommandations of all reviewers. I am therefore favorable to publication.

*Response: We would like to thank the Reviewer for his/her comments in every step of the peer review procedure. We feel that this constructive review provided significant aid in the improvement of the manuscript. The technical corrections proposed have been taken into account.*

Going through the manuscript, I noted few technical corrections:

- p2 l49 : "ED" is not defined

*Response: "ED" has now been changed to "emergency department"*

- p2 l52 : replace "particle" by "particulate"

*Response: Amended.*

- p3 l85 : replace "emerge" by "has become"

*Response: Amended.*

- p6 l174 : delete space character in "capt ure"

*Response: Amended.*

- p6 l185 : PSI is not defined

*Response: PSI is now defined.*

**REVIEWER #3**

The manuscript of Stavroulas et al. has been improved significantly after the revision. The interpretation of PMF factors is more solid now and all the details on the PMF analysis have been given in the supplemental material. I think that this manuscript merits publication after the minor comments listed below are taken into consideration.

*Response: We would like to thank the Reviewer for the detailed and thorough comments, which undoubtedly acted as a valuable guide towards the improvement of this manuscript.*

Minor comments:

1.      Manuscript needs careful proofreading, please double check figure numbers and abbreviations (they are spelled out several times or zero times at the moment)

*Response: The manuscript has gone through a thorough examination and proofreading. We believe all the mentioned accidental mistakes have been corrected in the revised version.*

2.      Abstract, "These results, based on the combined contribution of BBOA and SV-OOA, highlight the rising importance of biomass burning in urban environments during wintertime, as revealed through this characteristic example of Athens, Greece, where the economic recessions led to an abrupt shift to biomass burning for heating purposes in winter." This statement should be removed or toned down as this topic (increase in BB in last years) is not discussed in the manuscript anymore

*Response: The statement is toned down in the revised version of the manuscript.*

3.      Introduction is fragmented (short paragraphs on different topics), consider modifying

*Response: The introduction has been modified in the revised version taking into account the Reviewer's comment*

4.      Introduction; "to study the year-to-year changes of aerosol sources during winter time, with special emphasis on wood burning". This should be removed or modified as wood burning has not been investigated more extensively than other factors in the manuscript

*Response: This sentence has been modified in the revised version of the manuscript.*

5.      Page 18, line 550-551; COA concentration rising form (from?) 30% to 60% of the daily average during lunch hours (12:-15:00 LT). Please check these numbers, based on figures 5 and 6 these numbers can't be that large

*Response: The average daily concentration of COA during wintertime is 0.98 μg m-3. When calculating the average diurnal variability, we find that at 12:00 that the concentration is 0.3 μg m$^{-3}$, while at 15:00 it rises to 0.6 μg m$^{-3}$. These numbers correspond to the 30% and 60% respectively of the average daily concentration and have been added to the manuscript. Unfortunately, the steep rise in COA concentrations during night-time, creates a scale issue in the Figures.*

**Editor's comments on the revised version of ms. acp-2018-356 by I. Stavroulas et al., entitled "Identification of sources and processes that control submicron organic aerosol content at an urban Mediterranean environment using high temporal resolution chemical composition measurements"**

François Dulac, 16 Nov. 2018

I thank you for revision of your manuscript submitted to the ChArMEx special issue in ACP. I agree with both reviewers to acknowledge a great improvement in the manuscript. I invite you to follow their last recommendations for technical corrections and minor revisions. In the following, you fill find a list of additional technical corrections. Line numbers refer to the version of the manuscript with visible corrections included in your Authors' response. Changes are in red.

*Response: All the editor's additional technical corrections, reported in the following list, have been incorporated in the revised version of the manuscript. We would like to thank the editor for his thorough reading of the manuscript.*

- Abstract, line 30: round up to "22%".
- Introduction, line 86: not clear to me what you mean by "precursor emissions emerge altered".
- Instruments and Methods, line 126: add a comma after "Practically".
- Instruments and Methods, line 135: Budisulistiorini et al. (2014) is absent from the reference list; do you refer to doi:10.5194/amt-7-1929-2014?
- Instruments and Methods, line 181: "were used".
- Instruments and Methods, line 188: "were also calculated".
- Instruments and Methods, line 192: "to capture".
- Source Apportionment, line 207: "of factor profiles".
- Source Apportionment, line 225: I think you should explain what is ⍰ here, by reintroducing the sentence that has been erased from lines 235-236 ("The ⍰ value ranges between 0 and 1 and is a measure of"[…]).
- Source Apportionment, line 226: "Initial" or "Initially,".
- Choosing the optimal configuration, line 274: "On the other hand, constraining two factors during the warm period, namely".
- Choosing the optimal configuration, line 276: "previous".
- Choosing the optimal configuration, line 277: "during the warm periods".
- Chemical composition and charactristics, line 352: "while maxima".
- Chemical composition and charactristics, line 354: do you mean "(up to around 100"?
- Diurnal variability, line 403: "rush hours".
- Diurnal variability, line 423: "the organics variation also follows".
- Source apportionment of organic aerosol/Warm period, l.466: "solution stems from a two factor constrained run".
- Source apportionment of organic aerosol/Warm period, l.482: "can be attributed".
- Source apportionment of organic aerosol/Warm period, l.509: remove the comma before "(2005)".
- Source apportionment of organic aerosol/Warm period, l.517: "provide some".
- Source apportionment of organic aerosol/Warm period, l.551: "when concentrations".
- Source apportionment of organic aerosol/Warm period, l.559: "night-time".
- Source apportionment of organic aerosol/Warm period, lines 575-576: "and thus exhibits".
- Source apportionment of organic aerosol/Cold period, l.578: "solution stems from a three factor constrained run".

- Source apportionment of organic aerosol/Cold period, lines 607-608: "compared to BBOA in Bologna, Athens and Patras"; provide references.
- Source apportionment of organic aerosol/Cold period, l.621: by "extended area", do you rather mean "region"?
- Source apportionment of organic aerosol/Cold period, l.622: "The identification of BBOA […] as biomass burning tracers. Indeed, BBOA exhibits excellent correlation".
- Source apportionment of organic aerosol/Cold period, l.625: "and as reported".
- Source apportionment of organic aerosol/Cold period, l.630: do you mean "and exhibits" referring to BBOA, or "which exhibits" refering to BC$_{wb}$?
- Source apportionment of organic aerosol/Cold period, lines 636-637: decapitalize "northern and eastern Europe".
- Source apportionment of organic aerosol/Cold period, l.643: relove "who " and make a new sentence after "the basin": ". They found".
- Source apportionment of organic aerosol/Cold period, l.646: change "locality"; do you mean " "?
- Source apportionment of organic aerosol/Cold period, l.650: add a comma before "respectively".
- Source apportionment of organic aerosol/Cold period, l.661: "product fraction seems" (or "products seem" but pay attention to the following "Its" and "part of it").
- Source apportionment of organic aerosol/Cold period, l.661: do you mean oxidation of "combustion emissions" rather than "combustion sources"?
- Source apportionment of organic aerosol/Cold period, l.684: "SV-OOA".
- Source apportionment of organic aerosol/Cold period, l.691: check "1-hr lag the morning traffic."
- Source apportionment of organic aerosol/Cold period, l.692: do you mean oxidation of "combustion emissions" rather than "combustion source"?
- Summary and conclusions, l.720: "in addition to routine".
- Summary and conclusions, l.743: "peak values" or "values peaking".
- Summary and conclusions, l.755: "rush hours".
- Summary and conclusions, l.773: "contribution to air quality degradation"?
- References, l.802: year at the end.
- References, lines 808-809: "Meteor. Atmos. Phys."; remove issue number "(1-3)".
- References, lines 813-814: "Environ. Sci. Technol."; remove issue number "(15)"; final dot missing.
- References, l.841: "Meas. Tech.".
- References, lines 885-886: "Atmos. Environ.".
- References, lines 893-894: "emissions, Proc. Nat. Acad. Sci., 113, 10013"; add missing doi.
- References, lines 920 and 110: remove spare lines.
- References, lines 925-926: decapitalize a number of words in the article title; "Aerosol Sci. Technol.".
- References, l.1022: "Anal. Chim. Acta"; remove issue number "(1-2)".
- References, l.1026: "data values, Environmetrics"; remove issue number "(2)".
- References, lines 1067-1069: decapitalize a number of words in the article title; "Environ. Sci. Technol., 47, 13313-13320".

- References, l.1108: "Metorol., 124, 61-79" (no italic).
- References, lines 1112-1113: "Atmos. Environ." (dots).
- References, lines 1126-1128: "Fine particulate"; "Environ. Health, 8".
- Figure 6: put the left plot above and enlarge plots as in the previous version.
- Figure 7: put the left plot above and enlarge plots.
- Figure SF.21: if possible, use a different colour or lighter green for the land background in the map, to have more contrast with the green used for SV-OOA; enlarge the text boxes associated to each pie, and the numbers within the pies, or possibly turn the figure by 90° (landscape format) and enlarge.
* * *

[revised manuscript text omitted]